# Assessing Uncertainties in Modeling the Climate of the Siberian Frozen Soils by Contrasting CMIP6 and LS3MIP

Zhicheng Luo[1], Danny Risto[1], and Bodo Ahrens[1]

[1]Institute for Atmospheric and Environmental Sciences, Goethe University Frankfurt, Frankfurt a. M., Germany

**Correspondence:** Bodo Ahrens (Bodo.Ahrens@iau.uni-frankfurt.de)

**Abstract.** Climate models and their land components still show pervasive discrepancies in frozen soil simulations. Contrasting the historical runs of seven land-only models of the Land Surface, Snow, and Soil Moisture Model Intercomparison Project (LS3MIP) with their Coupled Model Intercomparison Project Phase 6 (CMIP6) counterparts allowed quantifying the contributions of the land surface parameterization scheme and the atmospheric forcing to the discrepancies. The simulation capabilities were assessed using observational data from 152 sites in Siberia and reanalysis data. In the winter months (December, January, and February), the LS3MIP ensemble bias in 0.2 m soil temperature was higher than the CMIP6 bias (-3.57 °C vs -2.66 °C). The spread of winter 0.2 m soil temperatures was also larger in the LS3MIP ensemble (4.55 °C) than in the CMIP6 ensemble (2.98 °C). For permafrost sites, the spatial correlation of the winter soil temperature simulations against observations was not better than 0.6, and the spring/autumn spatial correlations of snow depth were less than 0.8 for all CMIP6 models. On average, the 0.2 m soil temperatures in the CMIP6 simulations were 0.34 °C warmer than the observations when the simulated soil temperature dropped below -5 °C. However, the LS3MIP simulations were colder, with a bias of -0.65 °C. The biases of 2 m temperature had a different sign and were amplified in magnitude compared to the biases of the soil temperatures, especially below 0 °C. Four of the climate models and their land components underestimated the snow insulation effect. We concluded that land-only models have difficulties in accurately simulating soil temperatures and snow depth under low-temperature conditions. The CMIP6 models tended to compensate for errors in their land component with errors in the atmospheric model component. In shallow snow depth (0 to 0.2 m) cases, all models showed between 1 and 8 °C less air-soil temperature difference than in situ data. Therefore, a better representation of surface-soil insulation is essential for improvements in frozen soil land modeling.

## 1 Introduction

Under current climate conditions, an amplification effect up to 2–4 times is evident in warming trends within Arctic regions compared to global averages (Rantanen et al., 2022). Specifically, from 1979 to 2018, near-surface temperature growth rates in the land area surrounding the Arctic (0.51 °C per decade) were more than 1.5 times that of the Northern Hemisphere average

(0.33 °C per decade), according to Climate Research Unit (CRU) TS4.02 (Wang et al., 2022). Climate simulations indicate
that the Arctic experienced a warming rate of $0.66 \pm 0.32$ °C per decade from 1979 to 2014 (Cai et al., 2021). Additionally,
under a moderate scenario, high-latitude regions could see temperature increases ranging from 1.2 to 5.3 °C between 2005 and
2100 (Koven et al., 2013). Higher temperatures drive the degradation of permafrost, especially in discontinuous and sporadic
permafrost regions. The most significant changes occur in regions where the mean annual air temperatures are around 0 °C
(Romanovsky et al., 2007; Åkerman and Johansson, 2008; You et al., 2021). The soil temperature at zero annual amplitude
depth of continuous permafrost sites on a global scale warmed for $0.39 \pm 0.15$ °C during 2007–2016. This warming is two
to three times larger than that observed at discontinuous permafrost sites (Biskaborn et al., 2019; Smith et al., 2022), raising
concerns about the potential for more substantial permafrost degradation in the future.

Large amounts of soil carbon are stored in permafrost (Tarnocai et al., 2009; Fuchs et al., 2018), and when permafrost thaws,
the soil carbon could be emitted as carbon dioxide or methane into the atmosphere at a faster rate (Schädel et al., 2018). In
environments such as lakes and wetlands, the impact of thawed carbon on the climate is even more pronounced due to the
low-oxygen conditions, which further increases the proportion of methane emitted alongside other greenhouse gases (Koven
et al., 2015; Walter Anthony et al., 2018). Processes such as thermokarst result in sudden thaw events that greatly enhance the
decomposition and release of frozen soil carbon, potentially increasing carbon emissions by up to 50 % (Abbott and Jones,
2015; Turetsky et al., 2019). The winter contribution to total Arctic methane emissions is predicted to reach 39 % (Rößger
et al., 2022). Permafrost thaw alters hydrothermal conditions, which can alter surface vegetation depending on soil moisture. In
lowland regions with ice-rich permafrost, abrupt thawing is often followed by vegetation recovery. Under stronger or prolonged
changes, the system may reach a tipping point, beyond which widespread ecosystem disruption can occur (Heijmans et al.,
2022).

Despite its critical role in climate feedback processes, frozen soil remains one of the most uncertain components in Earth
system models. In the land surface component of climate models, heat transfer through the soil is typically simulated as one-
dimensional vertical transport. Models account for their specific soil layering schemes, where the thickness of soil layers
generally increases with depth. By calculating the water and thermal balance at different depths, land surface models can
derive the current state of soil moisture and temperature. Model parameters for hydrothermal transport are governed by soil
texture, surface organic matter, moisture dynamics, and freeze-thaw conditions (Woo, 2012; Andresen et al., 2020; Yang
et al., 2022). In permafrost regions, these factors determine the thermal offset (Kudryavtsev, V.A., 1977)—the temperature
difference between the ground surface and the top of the permafrost—by altering soil hydrothermal properties. For example,
the presence of solid and liquid water in frozen soil greatly affects the hydrothermal properties of the soil, which is described
in various ways by different models (Niu and Yang, 2006; Li et al., 2010). Incorporating soil ice and water dynamics into
land surface models improves simulations of active layer hydrothermal conditions by capturing seasonal freeze-thaw processes
and moisture effects, such as summer cooling and winter warming of permafrost due to increased soil moisture (Swenson
et al., 2012; Li et al., 2021; Du et al., 2023). Including soil ice dynamics in models allows for the simulation of ice-wedge
degradation and associated ground subsidence, capturing rapid landscape changes such as thermokarst under strong warming
scenarios (Liljedahl et al., 2016; Nitzbon et al., 2020; Cai et al., 2020). The parameterizations in land surface models are crucial

for accurately representing permafrost dynamics and their associated climate feedback processes (Yokohata et al., 2020), and they also determine the inclusion of key physical and biogeochemical processes essential for modeling permafrost carbon emissions (Ekici et al., 2015; Matthes et al., 2017, 2025).

The timescales of major physical processes differ largely between the soil and the atmosphere. For instance, key variables such as temperature and humidity in the near-surface atmosphere can experience substantial fluctuations within hours or even minutes. In contrast, changes in water and thermal states within the soil become much slower as depth increases. At depths of several tens of meters beneath permafrost, soil temperatures may not exhibit any noticeable variation over decades. In this context, it is essential to recognize that the soil surface serves as a critical interface for atmospheric interactions (Beringer et al., 2001; Langer et al., 2011a, b). Snow cover plays an important role by insulating soils while affecting surface energy balance through changes to local albedo as well as other characteristics such as emissivity and roughness. The impact of snow on soil temperature exhibits spatial heterogeneity based on snow's own attributes—specifically, thickness, density, and duration (Zhang, 2005; Zhang et al., 2018). A thick snowpack provides stronger thermal insulation, which limits soil heat loss in winter and delays thawing in spring. Lower-density snow insulates more effectively due to its reduced thermal conductivity. The duration of snow cover determines the length of the insulated period, which affects the timing and amplitude of seasonal soil temperature changes. Research has shown that changes in snow conditions (snow depth, density, and duration) account for over 50 % of variations in soil temperatures observed in northeastern Siberia (Park et al., 2014, 2015). An accurate representation of snow cover is essential for climate models, as a recent study has shown significant discrepancies in snow representations across different seasonal forecasting systems over Siberia due to different snow parameterizations and initialization methods (Risto et al., 2022).

The Coupled Model Intercomparison Project Phase 6 (CMIP6), launched by the World Climate Research Program (WCRP), aims to explore various topics related to climate change (Eyring et al., 2016). It allows evaluation of the ability of the latest generation of climate models to simulate frozen soil by providing an ensemble of climate models at resolutions fine enough to distinguish different frozen soil regions. The extent and characteristics of frozen soil can vary abruptly over short distances, especially in complex terrain or transition zones between different types of permafrost. Research efforts have been conducted to improve the simulation capabilities of land surface models participating in CMIP, focusing on biological and physical processes in frozen soil areas (Ekici et al., 2014; Chadburn et al., 2015; Decharme et al., 2016; Brunke et al., 2016; Jafarov and Schaefer, 2016; Guimberteau et al., 2018; Cuntz and Haverd, 2018; Damseaux et al., 2025). The Land Surface, Snow, and Soil Moisture Model Intercomparison Project (LS3MIP) is designed to enhance our comprehension of land surface processes by assessing the effectiveness of various land-only models in simulating soil temperature and moisture, snow cover, and related hydrological dynamics. It also aims to generate valuable insights that can aid in refining land-only models (Van Den Hurk et al., 2016). In LS3MIP, experiments are designed so that different land-only models use the same atmospheric forcing. Therefore, LS3MIP provides an opportunity to distinguish the impact of distinct climate variabilities produced by different atmospheric models in corresponding CMIP6 models.

In CMIP6 and LS3MIP, the setup of the land cover/land use scenario and the radiative forcing conditions follow the same protocol. However, the parameterization schemes of the climate models differ, and this is considered a main source of uncer-

tainty in climate modeling (Deng et al., 2021; de Vrese et al., 2023; Kuma et al., 2023). In addition, the presence of internal climate variability can also lead to differences in uncertainty (Ye, 2021; Rashid, 2021; Schwarzwald and Lenssen, 2022; Jain et al., 2023). Understanding and isolating these uncertainties is essential for improving model reliability.

Here, we focus on the Siberian region with frozen soils, which constitutes a significant portion of the Eurasian continent's frozen terrain. The potential degradation of permafrost in Siberia could have far-reaching consequences for climate and ecosystems throughout Eurasia and globally (Schuur et al., 2015; Streletskiy et al., 2025). Within this region, the observational dataset provided by the All-Russian Scientific Research Institute of Hydrometeorological Information-World Data Center (RIHMI-WDC) (Frauenfeld and Zhang, 2011; Sherstiukov, 2012a; Zhang et al., 2018) can be used. This dataset provides consistent soil temperature measurements at standardized depths and can thus be used as a reference in climate model evaluation.

The characteristics of frozen soil surface dynamics are assessed by comparing model outputs with references, including reanalysis and observational data. This research focuses on the shallow soil temperature response to atmospheric forcing, explicitly targeting a depth of 0.2 m. We will analyze the discrepancies between the climate models in CMIP6 and their land-only models in LS3MIP to quantify the bias and uncertainty present in frozen soil regions, attributing them to land-only models versus those resulting from atmospheric forcing. CMIP6 models include fully coupled components, such as the atmosphere, ocean, land, and sea ice, whereas LS3MIP models only include the land surface model and prescribe atmospheric forcing from reanalysis or historical simulations. Consequently, the differences between the two can be used to attribute model biases to either the land surface model structure and parameterizations or coupled atmosphere-land interactions. Under identical, observation-based atmospheric conditions, the LS3MIP models are expected to simulate soil temperature more accurately than their CMIP6 counterparts. If not, discrepancies may indicate limitations within the land surface models themselves, such as deficiencies in parameterization or missing processes that impair their ability to respond appropriately to atmospheric forcing. Conversely, errors found in coupled CMIP6 simulations may result from biases in atmospheric forcing, such as misrepresentation of near-surface air temperature, precipitation, or surface radiation. This experimental design allows us to distinguish the sources of uncertainty between land-only and coupled simulations. Additionally, we will explore inter-model variations within LS3MIP and assess how specific structural features, such as bottom boundary conditions and snow thermal conductivity parameterizations, relate to model performance in frozen soil regions.

## 2 Data and Methods

We used the data from climate models, reanalysis, observations, and processing methods of target variables for our analysis. We only included data from 1985 to 2014 in this research, as this period offers the best collection of observation records, and CMIP6 historical experiments are limited to 2014.

### 2.1 CMIP6 and LS3MIP Simulations

The CMIP6 multi-model ensemble provides historical climate simulations based on the same external forcing (solar radiation, greenhouse gases, aerosols, etc.) (Eyring et al., 2016). Our study used the *historical* simulations with predefined $CO_2$ concen-

**Table 1.** Selected CMIP6/LS3MIP experiment pairs, the layering and resolution. For other features and references, see Table 2.

| Model Name | Land Surface Model | Total Soil Layers (max. node depth (m)) | Soil Layers in Top 3 m | max. Snow Layers | Resolution (lat×lon) |
|---|---|---|---|---|---|
| CESM2 | CLM5.0 | 25 (42.0) | 14 | 12 | 0.9°×1.25° |
| CNRM-CM6.1 | Surfex 8.0c | 14 (10.0) | 11 | 10 | 1.4°×1.4° |
| CNRM-ESM2.1 | Surfex 8.0c | 14 (10.0) | 11 | 10 | 1.4°×1.4° |
| IPSL-CM6A-LR | ORCHIDEE v2.0 | 18 (65.56) | 12 | 3 | 1.25°×1.875° |
| HadGEM3-GC31-LL | JULES-HadGEM3-GL7.1 | 4 (2.0) | 4 | 3 | 1.25°×1.875° |
| UKESM1.0-LL | JULES-ES-1.0 | 4 (2.0) | 4 | 3 | 1.25°×1.875° |
| MIROC6 | MATSIRO6.0 | 6 (9.0) | 5 | 3 | 1.4°×1.4° |

trations, which contain distinctive combinations of atmospheric and land models. The *Land-Hist* experiments from LS3MIP are offline, land-only simulations with no feedback to the atmosphere and no dynamic forcing from atmospheric models (Van Den Hurk et al., 2016). All the LS3MIP simulations employed the same atmospheric forcing derived from the Global Soil Wetness Project Phase 3 (GSWP3) and the same land surface setup as in the CMIP6 experiments (Van Den Hurk et al., 2016).

GSWP3 is a global land surface modeling project that provides long-term meteorological gridded forcing data based on the 20th Century Reanalysis (20CR), bias-corrected with observational datasets. This setup allowed us to directly compare CMIP6 and LS3MIP results and disentangle the relative contributions of coupling-related errors and land model deficiencies to biases in frozen soil regions.

We chose seven climate models involved in both projects, incorporating six different land models. The selected models are

listed in Table 1. Other climate models, which also participated in both projects, could not be included in this study as they either turned off the freeze option in frozen soil in the CMIP6 version or did not provide data for all our target variables. Hereafter, we refer to the CMIP6 as Group C and the LS3MIP as Group L in plots and analysis. Four variables are collected for this study: 2 m air temperature ($tas$), soil temperature in 0.2 m depth ($tsl$), snow depth ($snd$), and precipitation ($pr$). Both CMIP6 and LS3MIP data can be accessed at https://aims2.llnl.gov/search/cmip6.

**Table 2.** Features of the land surface models. MC indicates mechanical compaction.

| Land Model | Snow Density | Snow Thermal Conductivity | Bottom Boundary Condition | References |
|---|---|---|---|---|
| CLM5.0 | MC | Density-dependent Jordan (1991) | Zero-flux | Van Kampenhout et al. (2017) Lawrence et al. (2019) |
| Surfex 8.0c | MC | Density-dependent Yen (1981) | Fixed temperature | Vionnet et al. (2012) Decharme et al. (2019) |
| ORCHIDEE v2.0 | MC | Depends on density, temperature, and pressure | Fixed flux | Wang et al. (2013) Bowring et al. (2019) |
| JULES-HadGEM3-GL7.1 | MC | Density-dependent Calonne et al. (2011) | Zero-flux | Clark et al. (2011) Walters et al. (2019) Wiltshire et al. (2020c) |
| JULES-ES-1.0 | MC | Density-dependent Calonne et al. (2011) | Zero-flux | Sellar et al. (2019) McNeall et al. (2024) |
| MATSIRO6.0 | Fixed $(300\,\mathrm{kg\,m^{-3}})$ | Fixed $(0.3\,\mathrm{W\,m^{-1}\,K^{-1}})$ | Fixed temperature | Takata et al. (2003) Guo et al. (2021) |

Table 2 highlights several attributes of each land model, focusing on their key characteristics related to the surface energy balance and associated processes. For a more general comparison of land-only models and their features regarding snow parameterization, see Menard et al. (2021). The land models differ in their representation of critical processes related to surface energy balance, snow physics, boundary conditions, and surface organic matter. The thermal conductivity of snow is modeled using density-dependent formulations as a power function (Yen, 1981) or a quadratic function (Jordan, 1991; Calonne et al., 145    2011; Wang et al., 2013), or using fixed values. Snow density is treated either dynamically through mechanical compaction (MC) or as a constant. The soil bottom boundary is handled with zero-flux assumption, fixed temperature, or fixed temperature gradients.

## 2.2 ERA5-Land Reanalysis

We utilized monthly averaged ERA5-Land reanalysis data from the European Centre for Medium-Range Weather Forecasts. ERA5-Land is a numerical land surface model product forced by atmospheric variables of ERA5, featuring a high horizontal resolution of $0.1° \times 0.1°$(Muñoz-Sabater et al., 2021; Copernicus Climate Change Service, 2019). Monthly data provided by ERA5-Land include 2 m temperature, snow depth, and soil temperature at four depths (0 to 0.07 m, 0.07 to 0.28 m, 0.28 to 1.0 m, 1.0 to 2.89 m). Soil temperature was linearly interpolated to a depth of 0.2 m to directly compare with observations. We also remapped ERA5-Land to a coarser horizontal resolution of 100 km (the finest resolution of the selected CMIP6 models). This remapped dataset is referred to as E5LC. Comparing ERA5-Land with its coarsened version allowed us to assess the impact of resolution on simulation outcomes. This also provided an opportunity to evaluate the reliability of ERA5-Land in representing conditions within permafrost-affected areas.

## 2.3 Observational Data

Observational daily data were gathered from 236 meteorology sites by RIHMI-WDC (Sherstiukov, 2012a; Bulygina et al., 2014a, b; Zhang et al., 2018). We filtered the data from 1985 to 2014 based on the quality flag (Sherstiukov, 2012b) provided in the dataset and employed only sites with a minimum of 330 valid days per year for all four target variables and at least 15 years of valid data. We only used the highest quality data, flagged as 0. Stations west of 60° E, east of 120° E, and south of 45° N were excluded to eliminate most stations with warmer climates. Moreover, sites were classified based on their winter 2 m air temperature. These criteria were put in place to ensure the accuracy and reliability of the data analyzed. A total of 152 stations were selected.

## 2.4 Data Preprocessing

We interpolated $tas$, $tsl$, $pr$, and $snd$ for all model simulations at the station sites' coordinates by choosing the nearest-neighbor grid cell. This introduced additional uncertainty when comparing modeled data with observed data. Notably, the CMIP6 historical runs can differ in phase from the actual climate phase due to internal climate variability. To minimize this uncertainty source, we considered only 30-year averaged data for evaluations.

## 2.5 Evaluation Metrics

To evaluate the models, we quantified their ability to simulate a reasonable climate mean state and internal climate variability.

The variability of target variables was quantified using the interquartile range (IQR), which measures the spread of the simulations. If the model over- or underestimated the observed $IQR_o$, the model over- or underestimated climate variability, respectively. Thus, we used the relative spread

$$RS_{m,i} = \frac{\text{IQR}_{m,i}}{\text{IQR}_{o,i}} \qquad (1)$$

with $m$ indicating the model, $o$ the observation, and $i$ the target variable.

The central climate state of the models and the observations were quantified by the median (med). We used the standardized model medians $\text{med}_{m,i}$, naming it relative bias $RB$,

$$RB_{m,i} = \frac{\text{med}_{m,i} - \text{med}_{o,i}}{\text{IQR}_{o,i}} \tag{2}$$

as a measure of systematic model errors.

$RS$ assesses a model's capability to reproduce the observed variability. This is essential for determining a model's reliability in simulating dynamic climate systems. $RB$, on the other hand, addresses systematic deviations using the median data values. It was standardized using observed variability ($IQR_o$), which facilitates comparisons of biases across different variables. $RB$ emphasizes whether systematic errors are pronounced relative to natural variability (informs if the error is smaller ($RB_{m,i} < 1$) or larger ($> 1$) compared to the observed climate variability), helping prioritize improvements in model development.

For the qualification of the error heterogeneity of the 2 multi-model ensembles at the sites' locations, we defined the ensemble means of the models' 30-year median biases as follows

$$EB_{i,s} = \sum_{m=1}^{M} \frac{1}{FN_m}(\text{med}_{m,i,s} - \text{med}_{o,i,s}) \tag{3}$$

where $s$ indicates the seasons, $F = 5$ represents the number of "model families" and $N_m$, which represents the number of "family members" (either 1 or 2 in this study), was used to properly weight similar models, as described in Kuma et al. (2023). We then calculated the ensemble spreads of median biases $ES_{i,s}$, which were defined as the standard deviations.

The ensemble mean biases $EB_{i,s}$ and spreads were calculated for both the CMIP6 (Group C) and LS3MIP (Group L) model ensembles with $M = 7$ members each. The analyses were done in different seasons to distinguish the impact of different freeze/thaw periods on ensemble performance. In this study, seasons were defined by months. Winter was defined as December, January, and February (DJF); spring as March, April, and May (MAM); summer as June, July, and August (JJA); and autumn as September, October, and November (SON).

## 2.6 Permafrost Stations

To investigate the climatology in the permafrost area, sites were identified as permafrost sites using the definition of Lawrence and Slater (2005). If a station had at least 300 days of valid data every year, and 24 consecutive months of lower than $0\,^\circ\text{C}$ in any layer below $0.2\,\text{m}$ depth, it was defined as a permafrost station. Using this method, we selected 19 permafrost stations among all observation sites.

## 3 Results

### 3.1 Winter 2 m Temperature in Target Area

Figure 1 shows the map of average winter (DJF from 1985 to 2014) 2 m air temperatures ($tas$) from the CMIP6 multi-model ensemble, and the symbols correspond to the observation data from RIHMI-WDC. As shown in the map, winter-time $tas$ in

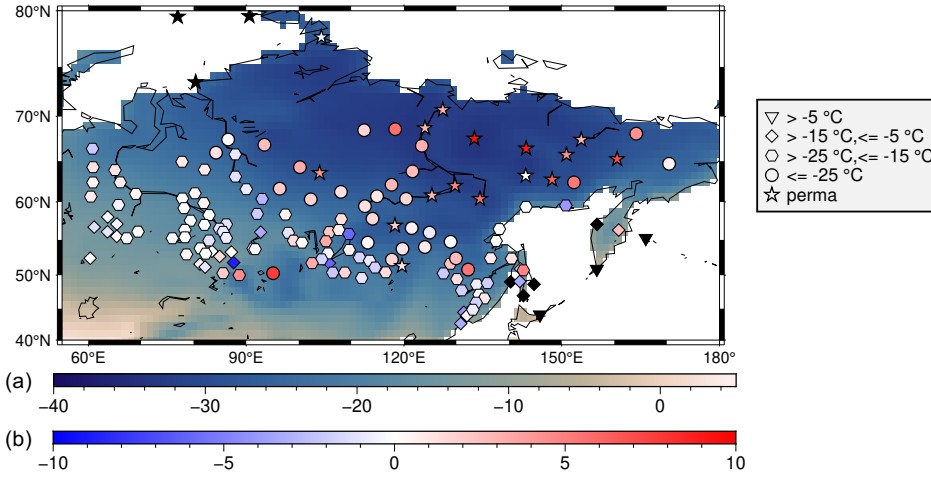

**Figure 1.** Mean near-surface air temperature (a) as given by the weighted multi-model ensemble from CMIP6, and the difference between the ensemble mean and the observational data (b) at sites for winter (DJF) in 1985–2014 in °C. Symbols indicate the climate state at the observational sites (see legend), and symbol colors show the difference between the ensemble mean and the observational data.

the target area is colder in the northeast and warmer in the southwest. Within the area 50° E to 185° E, north of 45° N, the average DJF $tas$ is generally below 0 °C. The region with less than -25 °C had a large overlap with the continuous permafrost region (Brown et al., 1997; Obu et al., 2019).

The temperature categories were listed in the legend of Fig. 1. There were 3 sites with average $tas$ warmer than -5 °C, 25 sites between -15 and -5 °C, 76 sites between -25 and -15 °C, and 29 stations with $tas$ below -25 °C. Besides, 19 sites were identified as permafrost ('perma') sites using the method introduced by Lawrence and Slater (2005) (soil layer temperatures continuously below 0 °C for at least two years) and labeled by stars. All 'perma' sites had average winter $tas$ values less than -23 °C.

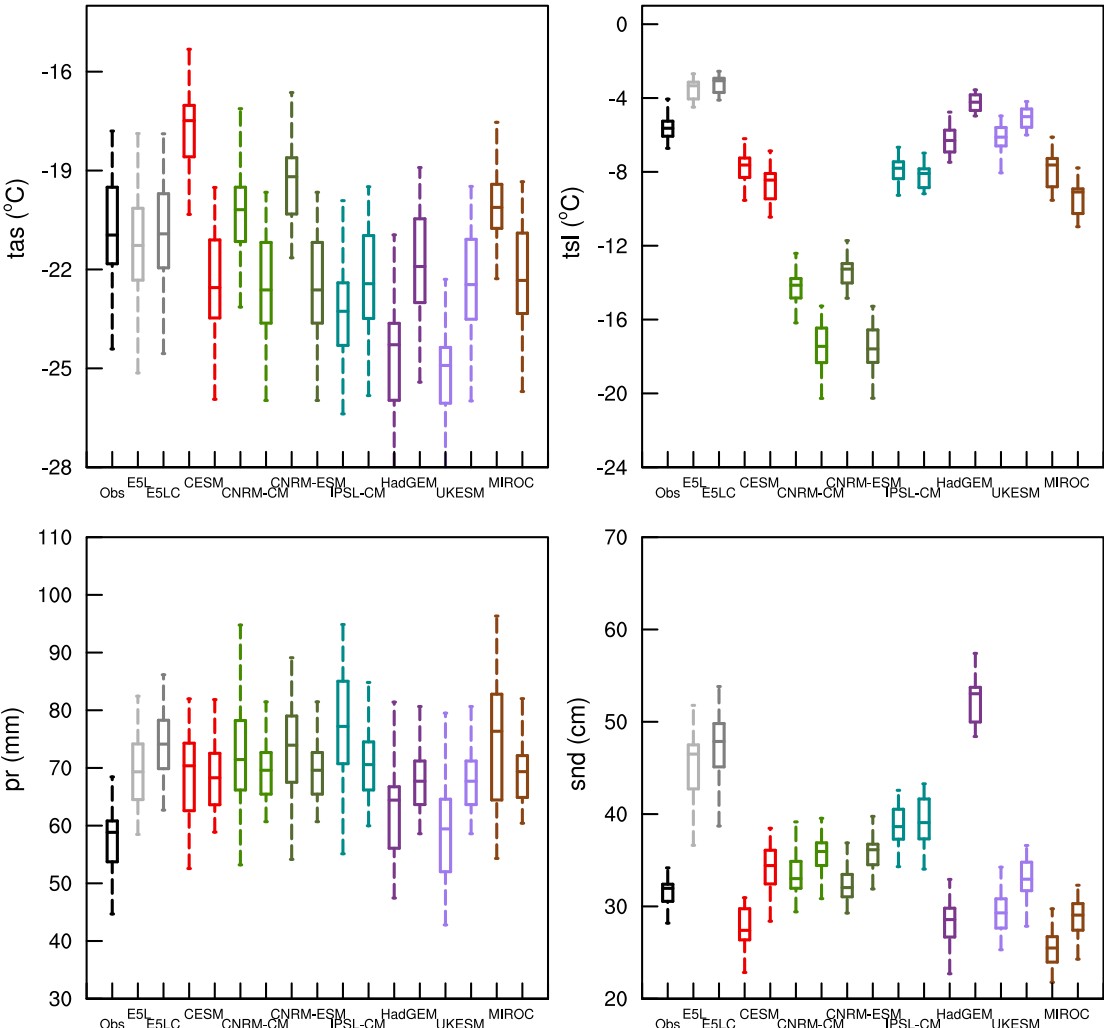

**Figure 2.** Sites' averaged DJF-climates (1985–2014) of hydrothermal variables as observed and simulated. Names on the x-axis and the colors indicate the different data sources. In addition to observations, ERA5-Land (E5L) and its coarsened variant, E5LC, are included to enable comparison at the same horizontal resolution as CMIP6 and LS3MIP. Model names indicate CMIP6 model output (left), and LS3MIP (right; see Table 1). Each CMIP6-LS3MIP pair shares the same color. The boxes represent the medians, first and third quartiles; the $\pm$ 1.5×IQR or the maximum and minimum values, if within the former range, are taken as the whiskers' length.

**3.2    Model Climatologies**

The sites' averaged winter climatologies (DJF) of the four target variables for the different datasets from 1985 to 2014 are shown in Fig. 2. All model variables were interpolated into the site's locations using the nearest neighbor method.

ERA5-Land's $tas$ and $tsl$ climatologies are closer to the observed climatologies than those of most models, though this is not the case for $pr$ and $snd$. Both ERA5-Land and E5LC were interpolated into the sites' locations using the nearest neighbor method. The coarser resolution resulted in slightly higher values of the four variables we analyzed, all less than one interquartile range of observations.

In LS3MIP, $tas$ and $pr$ were derived from the same forcing data. However, Fig. 2 shows slight discrepancies of $tas$ (less than 1 °C) and $pr$ (less than 3 mm) among land-only models. The LS3MIPs' $tas$ climatologies were systematically colder by more than 1 °C than observations and E5LC, and their $pr$ climatologies aligned better with E5LC than with observations (which have on average about 15 % smaller values).

The CMIP6 models' $tas$ climatologies scattered substantially. CESM2's $tas$ median was shifted by more than +3 °C against observations, while $tas$ medians of HadGEM3-GC31-LL and UKESM1.0-LL were shifted by more than -3 °C.

Most models' $tsl$ climatologies were colder than observed. The CNRM simulations exhibited the lowest soil temperatures overall (Fig. 2) with the climatologies of being more than 8 °C lower in both CMIP6 and LS3MIP. The strong cold bias of CNRM-CM6.1 and CNRM-ESM2.1 is also shown during spring and autumn (Fig. A2). As expected, the temporal variability, quantified by the box plots' IQR, was smaller in $tsl$ than in $tas$. Substantial inter-model variability was observed in soil temperature simulations. The differences between a model's CMIP6 and LS3MIP $tsl$ were much smaller than their differences from observations. Land-only models that belong to the same family (the land components of two CNRM models and the models HadGEM and UKESM, see Table 1) exhibited similarities in $tsl$ medians and IQRs.

In the other seasons, the spread of $tas$ and $tsl$ climatologies (Fig. A6) was respectively smaller than in DJF. During JJA, model differences in $tas$ variability were clearly reflected in $tsl$, particularly as snow was absent at most sites during summer (median value less than 0.3 cm).

In both CMIP6 and LS3MIP runs, the DJF $pr$ values were typically 10 mm higher than in the observations. UKESM1.0-LL simulated relatively good $pr$ in CMIP6, with less than 3 mm overestimation in DJF average. The observed DJF $snd$ was approximately 30 cm. It was overestimated by 70 % in HadGEM3-GC31-LL (LS3MIP run) and by 50 % in E5LC. The models were diverse in simulating DJF medians $snd$, with high temporal variability.

### 3.2.1 Relative Spread and Relative Bias

This subsection compares the relative spread ($RS$) and relative biases ($RB$) of E5LC and the different models compared to observations for all four target variables and all seasons. The $RS$ assesses the sites-averaged temporal climate variability, and $RB$ is the shift of the climatologies.

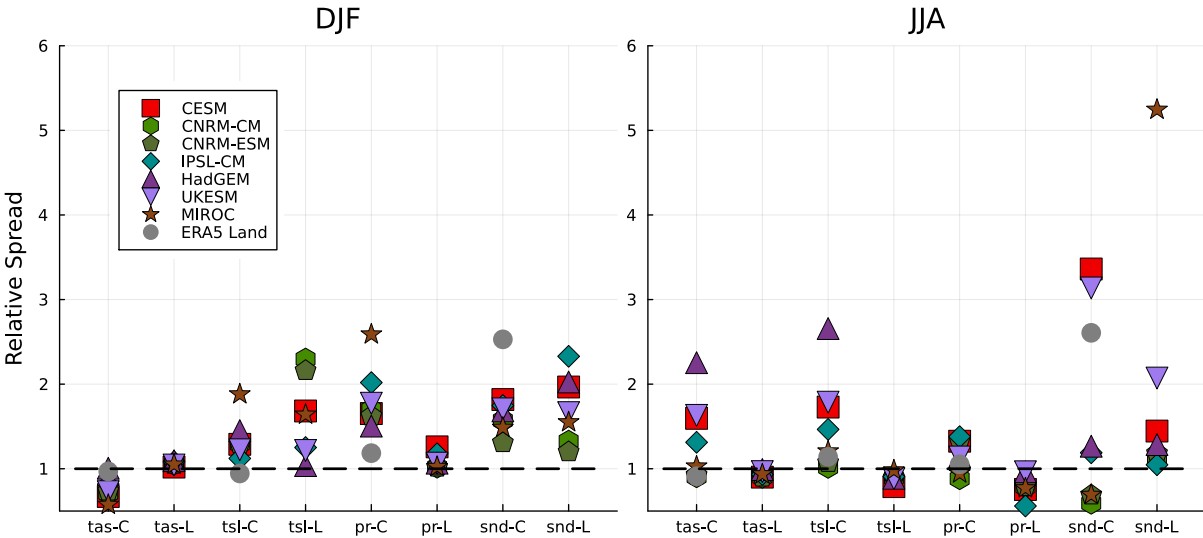

**Figure 3.** Relative spread ($RS$) of the sites-averaged climates (1985–2014) of the four variables in both ensembles and in E5LC with reference observations for all seasons. The colors indicate the models, and the x-axes show the variables and ensembles (C indicates CMIP6, and L indicates LS3MIP runs).

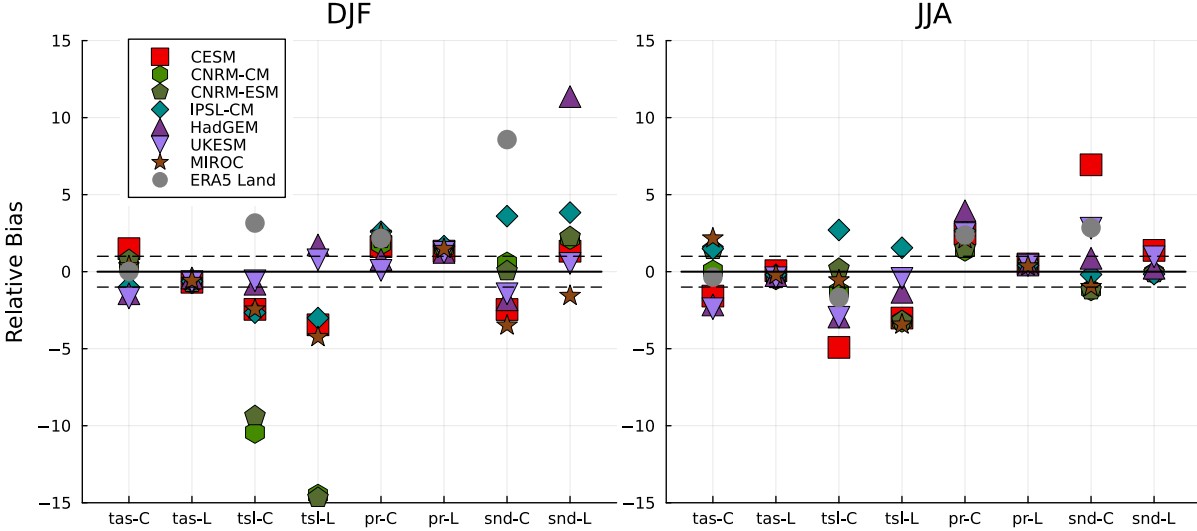

**Figure 4.** Same as in Fig. 3, but for relative biases (RB). The dashed lines (from -1 to 1) indicate the range of absolute median differences smaller than the observation's IQR.

Most CMIP6 models underestimated $tas$ DJF climate variability and models overestimated $tsl$ climate variability (Fig. 3). The CMIP6 models showed a larger diversity in $tas$, and all values were below 1. However, the LS3MIP models showed a large diversity for $tsl$ and $pr$ in winter, ranging from 1 to almost 2.7. The LS3MIP models performed well in simulating JJA variability of $tsl$, though there was a general underestimation. In DJF and SON, the $tas$ $RS$ for all CMIP6 models were within
the range of 0.5 to 1. In MAM, they were within the range of 0.7 to 1.7. CESM2, HadGEM3-GC31-LL, and UKESM1.0-LL exceeded 1.5 $RS$ of $tas$ in JJA. Larger variability differences in simulated $tsl$ were observed for LS3MIP simulations in DJF and SON, even though their $tas$ share almost identical $RS$.

     The DJF $pr$ relative spreads of Group C were all higher than 1.3. In spring and autumn, most climate models simulated good interannual variability of $pr$, with less than a 20 % difference to observation (Fig. A1). Both groups and E5LC overestimated
the spread in DJF $snd$, having high discrepancies among models. The $RS$ of $snd$ scatter between 0.8 and 1.8 in SON, and 1 and 1.6 in MAM for most models (except for E5LC, CNRM-CM and MIROC in CMIP6, and HadGEM in LS3MIP).

The relative bias $RB$ of all variables calculated in all seasons is shown in Fig. 4. The zone within the dashed lines indicates a one-time IQR of the observed value. If a model's absolute $RB$ is less than one, its performance is considered adequate. For example, the winter $RB$ of all LS3MIP models was within the dashed line zone for $tas$, and was at or slightly above 1 for the DJF $pr$ of all LS3MIP models as they are derived from the same atmospheric forcing data. Nearly all CMIP6 and LS3MIP models exhibited a positive relative $pr$ bias. The relative $snd$ bias was much more diverse; models only showed a consistent positive $snd$ bias in MAM (Fig A2). The CMIP6 runs of HadGEM3-GC31-LL and UKESM1.0-LL underestimated the values of $tas$ and $tsl$ in all seasons. The $tsl$ $RB$ in DJF for CNRM simulations exceeded -15 to -9 times the observed IQR, while other models were all within a range of $\pm 5$ times in both groups. The $RB$ of CMIP6 and LS3MIP in $tsl$ were all negative in the transition seasons, and also high discrepancies were shown in $snd$ $RB$ values.

### 3.2.2 Spatial Heterogeneity

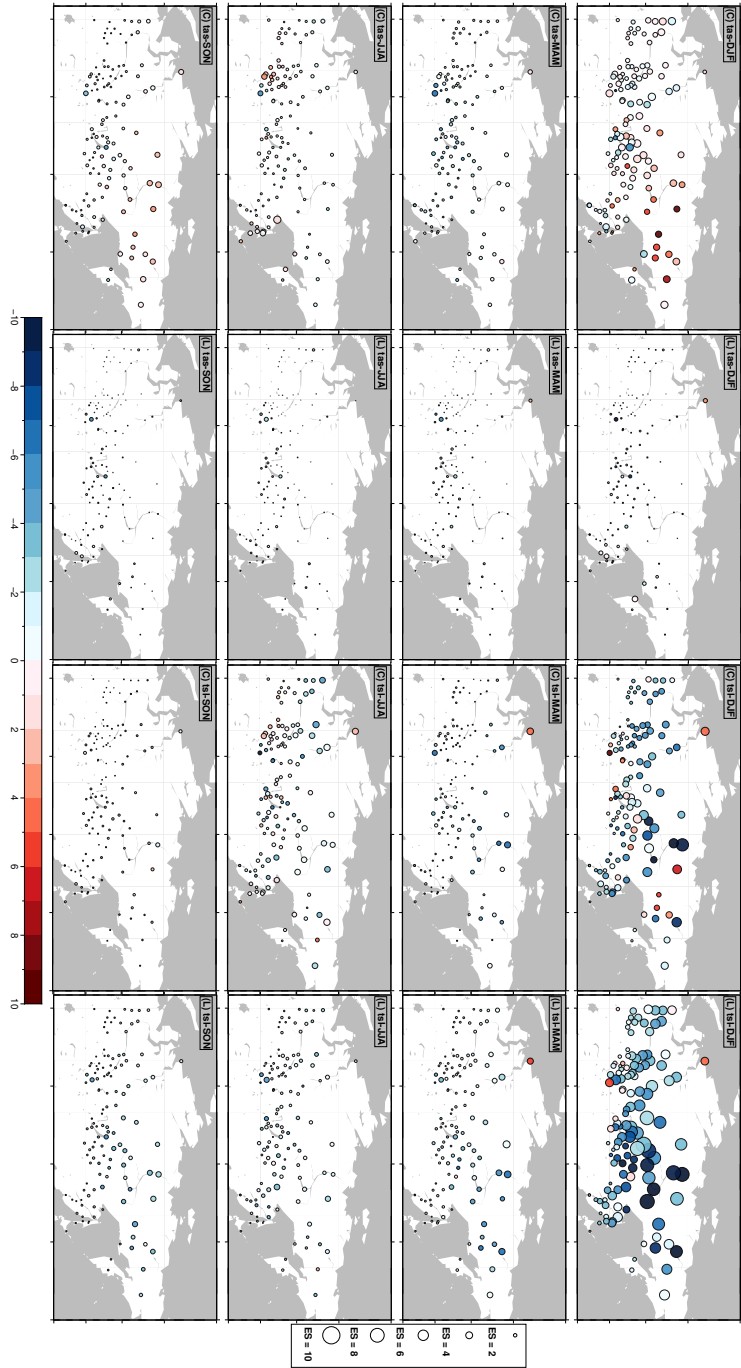

**Figure 5.** Weighted multi-model mean biases ($EB$) and their standard deviations ($ES$) of $tas$ and $tsl$ at observational sites in °C. Panels A–H and I–P are for $tas$ and $tsl$, respectively. The first and third columns are the results of CMIP6 models, and the second and fourth columns are the results of LS3MIP models. The first to fourth rows show the DJF, MAM, JJA, and SON seasons, respectively. Colors indicate the bias value, while the circle radii represent biases' standard deviations, i.e., the ensemble spread.

In DJF, multi-model ensembles had the largest $EB$ and $ES$ (Fig 5). In DJF, the CMIP6 models overestimated $tas$ on average by more than 0.5 °C, but with a large $ES$ of 2.96 °C. The DJF average $EB$ and $ES$ for LS3MIP were -0.51 °C and 0.69 °C, respectively.

In the other seasons, $EB$ and $ES$ were distinctly smaller, with most CMIP6 runs' $tas$ slightly too cold in spring at most stations and slightly too warm in autumn in northeastern Siberia. The $tas$ $EB$ of the forced models was small at most sites (exceptions were near water bodies, e.g., Lake Baikal and sea coasts, which indicated interpolation artifacts due to the selected grids).

The $tsl$ $EB$ and $ES$ were larger in magnitude than for $tas$, especially in winter, with spatial averaged $EB$ of -2.66 °C and 275     -3.57 °C for CMIP6 and LS3MIP runs, respectively. Additionally, the LS3MIP runs had a larger bias spread than the CMIP6 runs in all seasons except summer. In winter, the $ES$ were 2.98 °C for the CMIP6 and as large as 4.55 °C for the LS3MIP runs.

## 3.3  Permafrost Region

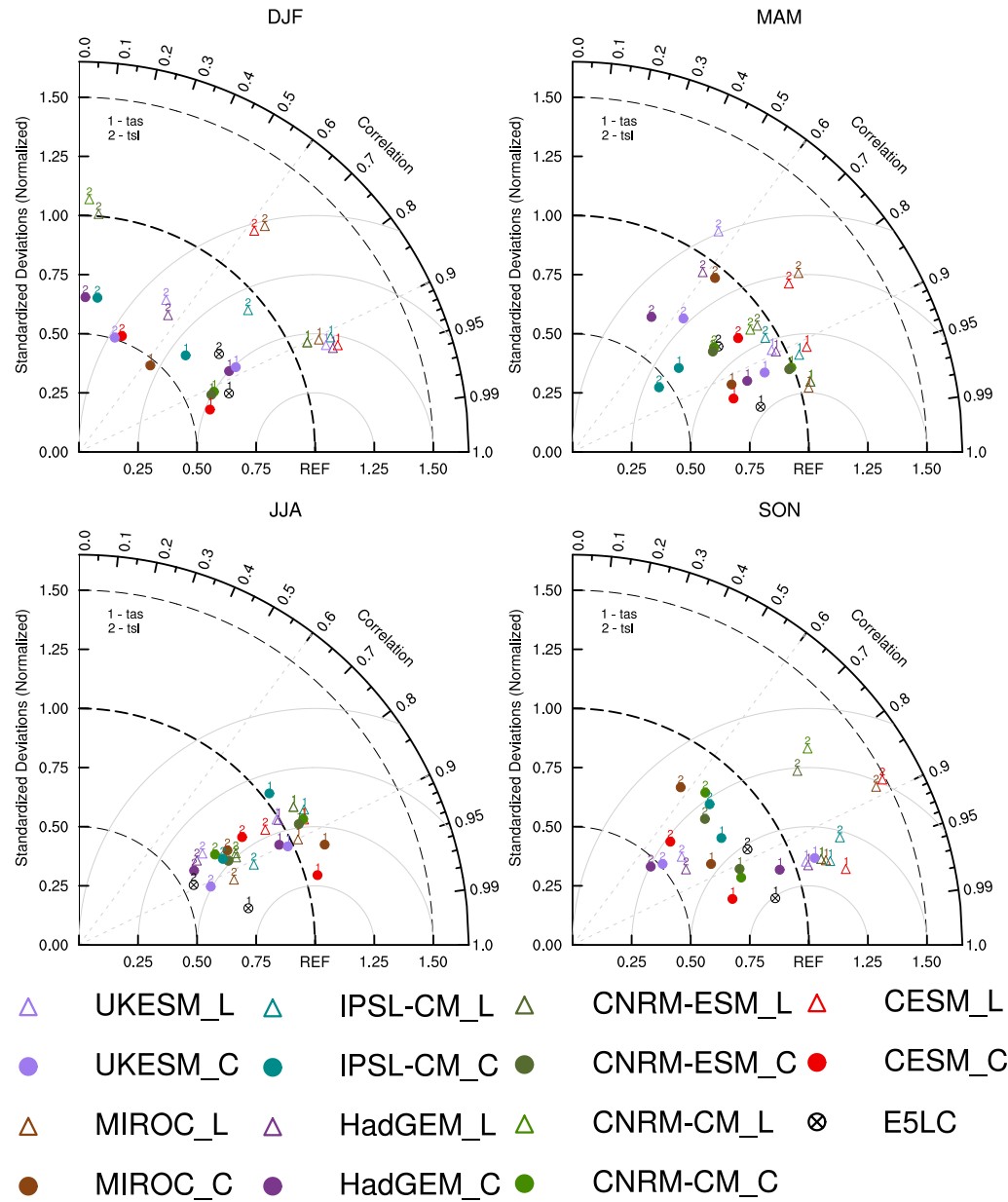

**Figure 6.** Taylor diagrams indicating the spatial correlation, standard deviation, and unbiased root-mean-square (RMS) difference (gray circles) for the four seasons of the simulated variables $tas$, $tsl$ (labeled by numbers) against observations (REF) at permafrost sites (here: sites with mean winter $tas < -25\,^{\circ}\mathrm{C}$). Normalization is applied using the standard deviations of the observations. Colors indicate different models and black crossed circles E5LC. Triangles show the results of the LS3MIP runs, and solid circles of the CMIP6 runs.

Using the classic definition of permafrost, which requires more than two consecutive years of temperatures below $0\,°C$, 19 permafrost sites were selected. Notice that in the permafrost region, the largest $EB$ and $ES$ are shown (see Fig. 5). The simulation performance at the permafrost sites was quantified using Taylor diagrams (Taylor, 2001). The 30-year average of simulated data was compared to the observations in all seasons at all sites in the permafrost region. The REF in Fig. 6 is the corresponding observation.

In JJA, both ensembles showed a high correlation with the observations (higher than 0.7). Both ensembles simulated JJA $tas$ within a standard deviation ranging from 0.95 to 1.1, while E5LC's standard deviation was lower than 0.75. In the other seasons, the $tas$ simulations had correlations higher than 0.6; however, they all underestimated the standard deviation of $tas$. In MAM and SON, $tas$ correlations were almost above 0.9, which was slightly higher than $tas$ correlations in JJA.

Almost all CMIP6 and LS3MIP models struggled to simulate $tsl$ in permafrost sites, exhibiting low correlations and large RMSE, especially during DJF. The correlations were generally higher than 0.5 for all variables in the other seasons. E5LC had similar DJF $tas$ simulation performance to the CMIP6 models, but it simulated DJF $tsl$ with the smallest RMSE compared to all models. When considering the same model across CMIP6 and LS3MIP, land-only simulations consistently outperformed others in DJF $tsl$. This is because $tas$ and $snd$ were superior in LS3MIP (Fig. A3).

From spring to autumn, the CMIP6 models tended to overestimate precipitation variability (see Fig. A3). Overall, the Group L models and E5LC performed better than the Group C models in the MAM and SON $snd$ simulations. In MAM, the standard deviation of $snd$ was considerably larger than in DJF and SON, exceeding 1.25 for Group C models and 1.5 for Group L models. Using the same forcing, the $snd$ correlations improved in SON, with all values above 0.8 except for UKESM1.0-LL, which had a correlation of 0.7. The correlations of CMIP6 $pr$ were similarly low in all seasons. Despite LS3MIP showing a high correlation (higher than 0.8) and good standard deviation (0.9 to 1.1) of $pr$ in DJF and MAM, CMIP6 and LS3MIP were unable to simulate $snd$ correlations higher than 0.8 in these two seasons.

## 3.4 Climate Dependency of Modeled Temperatures

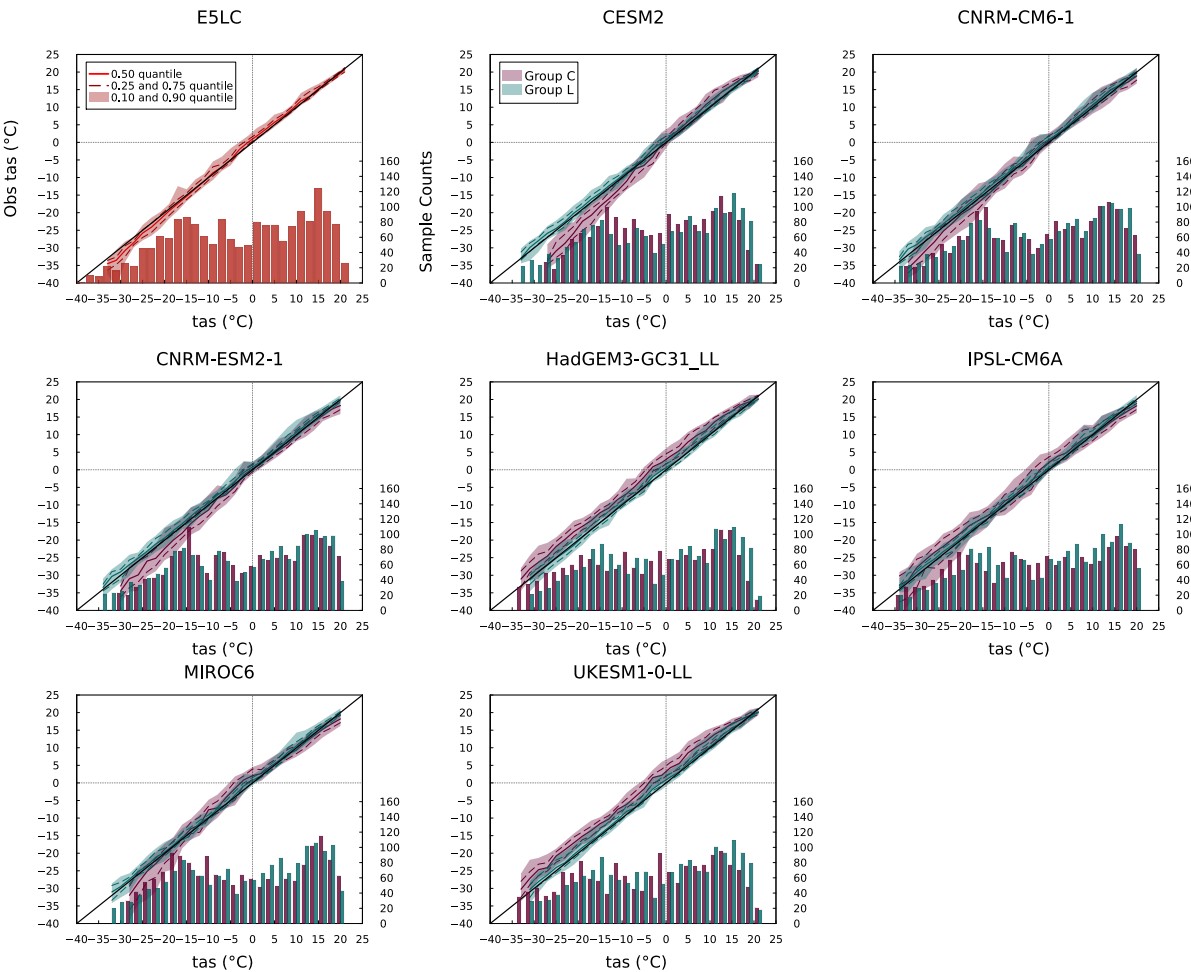

**Figure 7.** Quantile-Quantile (QQ) and histogram plots for $tas$ (Wilks, 2019). The plots show 30-year mean monthly $tas$ data of models at all sites and their observational values. The model simulation outputs are binned at $2\,^{\circ}\text{C}$ intervals. The colored solid and dash curves are the median and 1st/3rd quartiles of the corresponding observations for all data points in the temperature interval, and the shaded area is the inter-decile range. The histograms represent the sample size within each temperature interval, and temperature intervals with sample sizes smaller than 20 are excluded from the Q-Q plots. The further away the data is from the diagonal line, the larger the model's simulation bias is at certain temperature states. A higher vertical quantile range indicates more inconsistency with observation under identical temperature conditions.

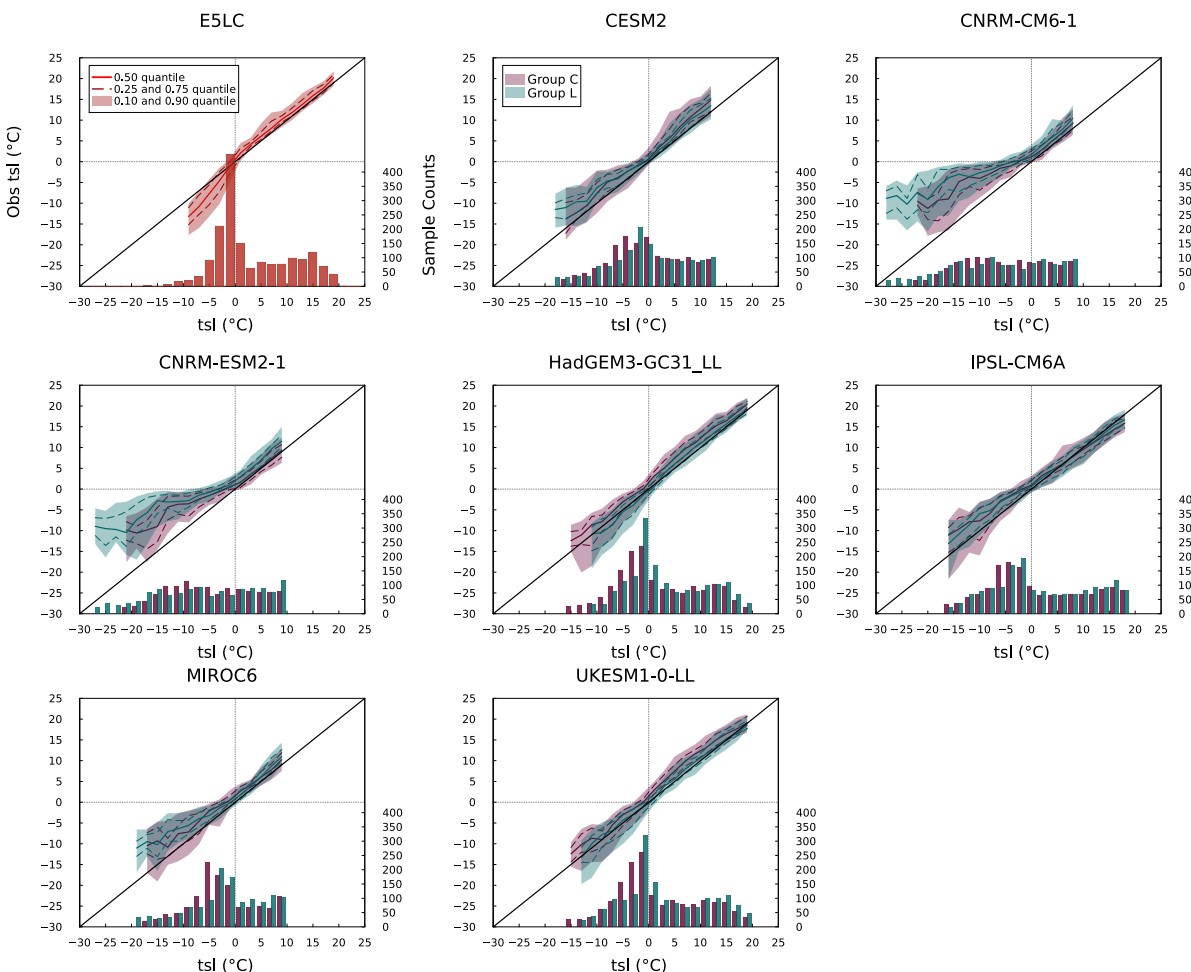

**Figure 8.** Same as Fig. 7, but for tsl.

Larger model biases and discrepancies were found in permafrost regions. To evaluate how model errors depend on the temperature conditions, we categorized all the model outputs of $tas$ and $tsl$ based on their values and compared these outputs to the corresponding observations. For each model, the monthly data from every site was calculated as an average over 30 years. We then examined the corresponding values for the same site and month with valid observations, calculating the median values, quartiles, and deciles for each interval.

As shown in Fig. 7, Group L models generally reproduced the observed sample distribution of $tas$ and maintained narrow quantile spreads across the full temperature range. In contrast, Group C models exhibited larger deviations. For example, CESM2, CNRM-CM6.1, and CNRM-ESM2.1 tended to simulate $tas$ higher than observed when the observed $tas$ was below 0,°C, with median differences exceeding +3 °C. HadGEM3-GC31-LL and UKESM1.0-LL, by contrast, consistently produced $tas$ values approximately 5 °C lower than observed across a broad range of frozen conditions. IPSL-CM6A-LR showed per-310 sistent cold biases of over -5 °C in most subzero-temperature cases. MIROC6 produced $tas$ values that were lower than the observed values between -10 and 2 °C, but the simulated $tas$ values were increasingly higher than the observed values when simulated $tas$ dropped below -20 °C. A common feature of Group C models was a wider vertical spread of quantiles in the below-freezing temperature range, especially when $tas$ dropped below -15 °C.

    Compared to $tas$, quantile spreads were generally larger in the $tsl$ simulations. For instance, the E5LC model exhibited a 315 bimodal $tas$ sample distribution (Fig. 7), with one peak near 15 °C and another around -15 °C, whereas its $tsl$ sample distribution (Fig. 8) showed a single dominant peak between -5 and 2 °C. E5LC also displayed smaller and more centered $tsl$ quantiles above 0 °C, but it showed an increasing warm bias and quantile spread as $tsl$ dropped below freezing. Similar near-surface soil warming was observed in the LS3MIP run of HadGEM3-GC31-LL, which was associated with excessive snow depth. CNRM-CM6.1 and CNRM-ESM2.1 were the only models that did not show a cluster of $tsl$ values near 0 °C. Instead, their 320 $tsl$ sample distributions were shifted toward much lower values (with extreme values lower than -15 °C), showing median differences from observations reaching -17 °C. CESM2 and MIROC6 also displayed systematic cold biases, with median $tsl$ differences of -5 and -6 °C, respectively, when the observed $tsl$ was around -10 °C. Group L models varied more in magnitude but generally showed positive $tsl$ biases below 0 °C.

    The two groups also showed distinct differences in simulating cold temperature extremes, particularly in the minimum $tas$. 325 For example, CESM2's minimum $tas$ in the coupled simulation was 6 °C higher than in its land-only simulation, indicating a notable warm bias in the coupled system at the lower extreme. In contrast, HadGEM3-GC31-LL and UKESM1.0-LL simulated minimum $tas$ values that were 2 °C lower in the coupled runs than in their land-only counterparts. These discrepancies in $tas$ at the cold end of the sample distribution were also reflected in the $tsl$ simulations. In particular, the minimum $tsl$ values differed by up to $\pm 6$ °C between models in the two groups.

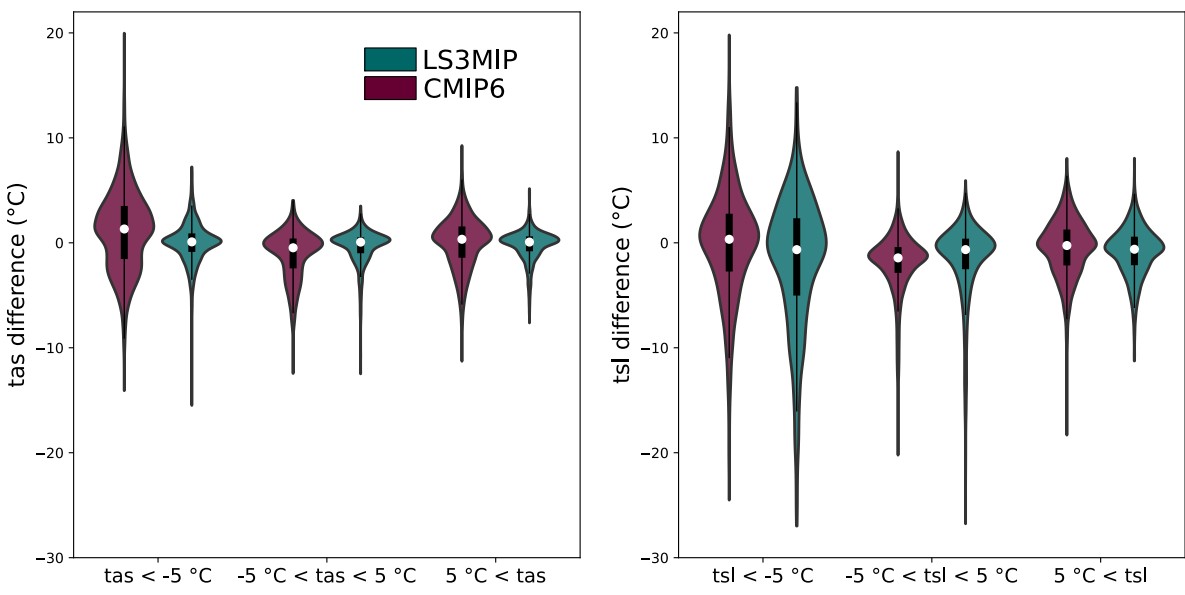

**Figure 9.** Difference between model $tas$ (left), $tsl$ (right) and corresponding observations. The x-axis represents different temperature intervals, and the y-axis is the 30-year average temperature difference ($T_{model} - T_{obs}$). Differences are categorized into three sets according to the 30-year average temperatures of every month at the observation sites: below -5 °C (Set Frozen), -5 °C to 5 °C (Set Intermediate), and above 5 °C (Set Warm). The violin plots show the distribution of the data. The width represents the density of the data points. The white dots show the median values, and the thick vertical black lines show the interquartile range.

**Table 3.** Statistics of the differences between the weighted multi-model ensembles and the observations of $tas$ and $tsl$, as illustrated in Fig. 9.

| Category | Sample Size | Ensemble | $Q_1$ | Median | $Q_3$ | Mean | Std. Dev. | Kurtosis |
|---|---|---|---|---|---|---|---|---|
| $tas$ Set Warm | 3475 | CMIP6 | -1.42 | 0.34 | 1.56 | 0.01 | 2.46 | 0.76 |
| | | LS3MIP | -0.81 | 0.07 | 0.63 | -0.22 | 1.43 | 2.93 |
| $tas$ Set Intermediate | 1480 | CMIP6 | -2.44 | -0.47 | 0.41 | -1.12 | 2.28 | 0.77 |
| | | LS3MIP | -0.99 | 0.06 | 0.52 | -0.38 | 1.61 | 7.79 |
| $tas$ Set Frozen | 3805 | CMIP6 | -1.55 | 1.32 | 3.51 | 1.16 | 3.85 | 0.59 |
| | | LS3MIP | -0.87 | 0.09 | 0.90 | -0.04 | 2.06 | 6.64 |
| $tsl$ Set Warm | 3295 | CMIP6 | -2.15 | -0.26 | 1.26 | -0.51 | 2.81 | 1.60 |
| | | LS3MIP | -2.13 | -0.61 | 0.58 | -0.86 | 2.35 | 1.21 |
| $tsl$ Set Intermediate | 3655 | CMIP6 | -2.86 | -1.44 | -0.41 | -1.95 | 2.87 | 4.33 |
| | | LS3MIP | -2.51 | -0.65 | 0.40 | -1.70 | 3.81 | 6.41 |
| $tsl$ Set Frozen | 1445 | CMIP6 | -2.74 | 0.34 | 2.77 | -0.12 | 5.26 | 1.68 |
| | | LS3MIP | -5.03 | -0.65 | 2.34 | -1.81 | 6.43 | 1.24 |

We subtracted the 30-year average of monthly observational data from all stations with the corresponding simulated values and then sorted them into observed temperature intervals, as shown in Fig. 9. The boundary of -5 °C and 5 °C was chosen to make sure the soil is completely frozen/thawed in the Set Froze/Set Warm.

Statistical values of Fig. 9 were listed in Table 3. Set Frozen and Set Warm $tas$ data sample sizes were more than twice as large as in Set Intermediate. Set Frozen had the largest standard deviations, 3.85 °C in CMIP6 runs, and 2.06 °C in LS3MIP

runs. The mean values of the CMIP6 runs were divided, with -1.12 °C and 0.01 °C in Set Intermediate and Set Warm, respectively.

The $tsl$ samples were mainly concentrated in Set Intermediate and Set Warm. In contrast, there were also higher standard deviations in Set Frozen, 5.26 °C and 6.43 °C for CMIP6 and LS3MIP runs, respectively. The standard deviations of Set Frozen and Set Intermediate in LS3MIP runs were higher than those in CMIP6 runs by 1.18 °C and 0.94 °C, respectively. The mean

and minimum value of $tsl$ bias was much lower than that of $tas$ bias, and this negative bias was shown in all corresponding sets of both groups in Table 3. And $tsl$ below -5 °C showed higher overall variability in Group L than in Group C, with the highest IQR of 7.37 °C, and highest standard deviation of 6.43 °C among all categories.

### 3.5    Snow Insulation

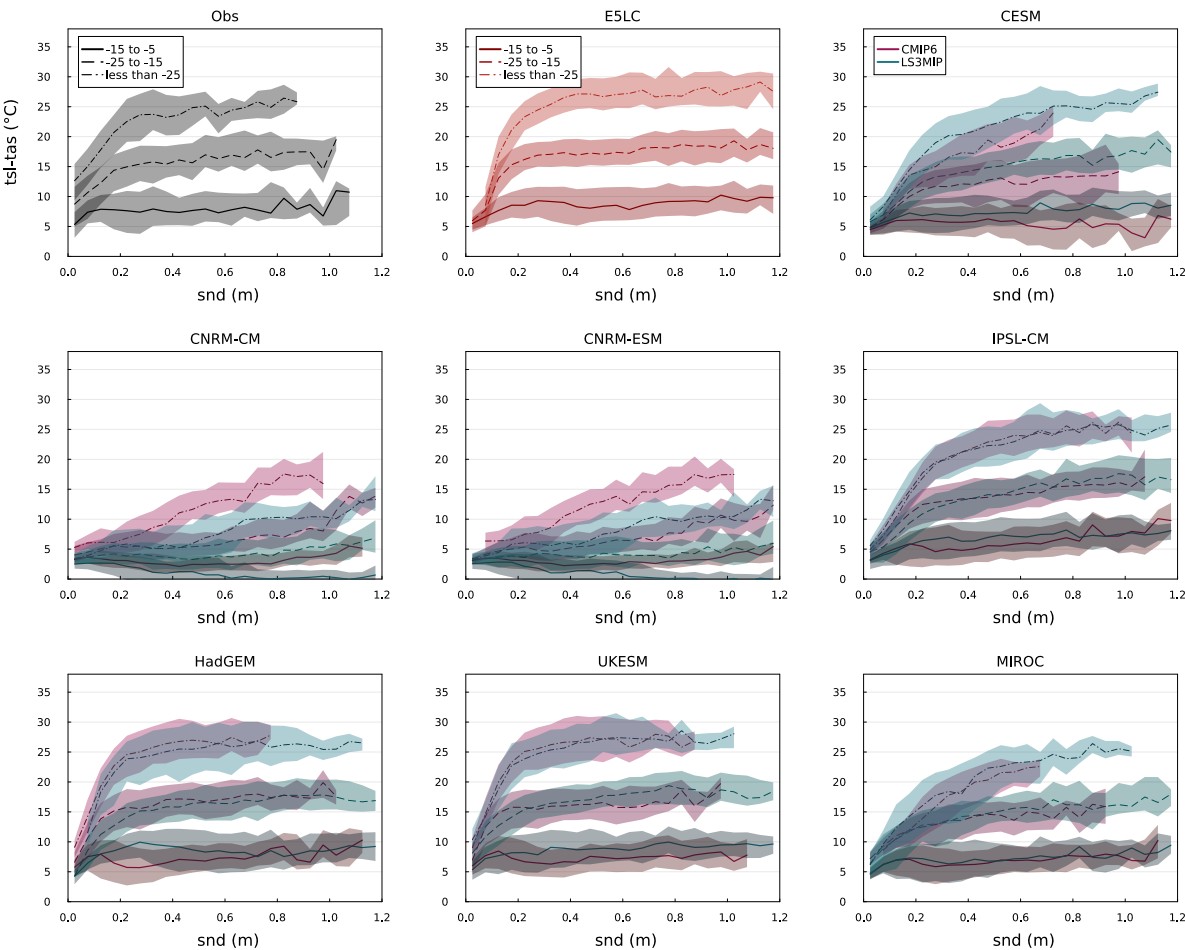

**Figure 10.** Snow depth and soil-surface temperature differences. The temperature categorization method follows Wang et al. (2016) but includes data from all seasons. The plots use different line styles and color schemes to distinguish between different temperature ($tas$) categories and ensembles (CMIP6 in red and LS3MIP in blue). Our sampling technique involves binning data at $0.05\,\mathrm{m}$ intervals, with coverage of 0–1.2 m $snd$, and each interval contains a minimum of ten samples for accuracy. The lines represent the median values, while the shaded areas indicate the interquartile range (25th–75th percentile).

To better understand the insulating effect of snow in the models, we applied a method similar to that used in previous model evaluations (Wang et al., 2016; Burke et al., 2020). We collected the monthly observational data samples containing $tas$, $tsl$ and $snd$ and characterized them by three different $tas$ intervals (-15 to -5 °C, -25 to -15 °C, and less than -25 °C). Analyzing the relationship between temperature difference and snow depth allowed us to understand each model's snow insulation effect under different $tas$ and $snd$ conditions.

According to the observations shown in Fig. 10, even when $snd$ is very thin, the insulating effect of the soil surface layer was still large. Its intensity could range from 3 °C to 15 °C according to $tas$. The impact of snow on $\Delta T$ ($tsl - tas$) showed the greatest variability when the snow depth is shallow. However, once the snow depth exceeded about 0.2 m, $tsl$ stabilized near 0 °C or lower, becoming nearly constant and minimally affected by $tas$. For temperatures ranging from -15 to -5 °C, this inflection point was reached when $snd$ was about 0.1 m high. For colder temperatures, it was reached at depths around 0.2 m to 0.3 m. Thicker snow conditions limited the cooling of the underlying soils, as indicated by the strong relationship observed between $\Delta T$ and $tas$ when $snd$ exceeded $0.3 m$. The median value of maximum $\Delta T$ for the -5 °C to -15 °C category was approximately 7.5 °C, while for the -15 °C to -25 °C category, it was 15 °C, and it reached 24 °C when $tas$ was below -25 °C. When $snd$ exceeded 0.3 m, $tsl$ drops by 1 °C for every 4 °C to 5 °C decreased in $tas$, considering the range of interquartile variations. E5LC showed similar insulation in the two warmer temperature categories. However, it showed lower insulation when there was almost no snow and generally larger insulation in the coldest category.

Only HadGEM3-GC31-LL, IPSL-CM6A-LR, and UKESM1.0-LL exhibited $\Delta T$ curves that were similar to observations. Despite the generally good performance, HadGEM3-GC31-LL and UKESM1.0-LL had deficiencies in simulating soil insulation below -15 °C, with $\Delta T$ 1 to 2 °C higher than observation when $snd$ was higher than 0.3 m. IPSL-CM6A-LR also underestimated the insulating effect when there was almost no snow. However, it accurately reproduced the inflection point of the $snd$-$\Delta T$ curve. After reaching this point, the insulation effect still continued to increase in IPSL-CM6A-LR. In contrast, HadGEM3-GC31-LL and UKESM1.0-LL showed a slower rate of increasing $\Delta T$ that was more comparable with the observations. The other models tended to underestimate the snow insulation, with the LS3MIP simulations from CESM showing results closer to observations, while the CNRM simulations showed the largest underestimation.

Comparing the CMIP6 runs with the corresponding LS3MIP runs, UKESM1.0-LL, IPSL-CM6A-LR, and MIROC6 showed similar snow insulation effects in both ensembles (CMIP6 ensemble and LS3MIP ensemble). HadGEM3-GC31-LL also showed comparable characteristics, although it differed under warmer conditions with lower snow depths. CESM2's LS3MIP runs showed a snow insulation effect closer to observations, but CESM2's CMIP6 runs simulated a much lower $\Delta T$ curve. In the LS3MIP CESM2 runs, under the colder conditions (i.e., below -15 °C), the snow insulation effect increased consistently with snow depth over all depths. Similar results were observed in MIROC6 as well. Both models showed realistic snow insulation effect values when $snd$ is above 0.4 m. However, the models' initial insulation started low and increases more slowly in $\Delta T$ when $snd$ was below 0.2 m. This was even more pronounced in CNRM-CM6.1 and CNRM-ESM2.1, which had a low $\Delta T$-$snd$ curve and a negative $\Delta T$ value for $snd$ above 0.2 m under the -5 °C to -15 °C category of LS3MIP runs.

## 4 Discussion

### 4.1 Winter 2 m Temperature in Target Area

We aimed to assess the models' performance under varying climate conditions to determine whether simulation uncertainties increase at lower temperatures or remain similar. To distinguish different climate regimes, a practical approach to categorizing the stations is to use their average DJF 2 m air temperature, following Wang et al. (2016). By focusing on winter temperatures, we could further link the results to the insulating effect of snow.

### 4.2 Model Climatologies

Horizontal resolution did not have a major contribution to bias, concerning differences between ERA5-Land and E5LC in Fig. 2. Even though the LS3MIP models were forced with identical reanalysis data, their $tas$ and $pr$ medians differed slightly, as they have different grid resolutions and methods for calculating and regridding $tas$ and $pr$. Also, the LS3MIP $tas$ and $pr$ differed from observations, highlighting the fact that a site's climate does not necessarily represent the climate of the corresponding grid. This emphasizes the importance of recognizing that gridded model outputs are spatial averages constrained by grid cell resolutions. Station observations, on the other hand, may not represent the broader grid-scale climate. Therefore, higher consistency between the two does not necessarily imply better model performance. Furthermore, discrepancies in temporal variability between modeled and observed climate can complicate interpretation when comparing with CMIP6 outputs. In this study, we focused on evaluating the models' ability to simulate $snd$ and $tsl$ with similar atmospheric forcing ($tas$ and $pr$). The cold bias in LS3MIP $tas$ climatologies and their closer $pr$ agreement with ERA5-Land than with observations suggested shared biases from reanalysis and model forcing, which was also possibly due to differences between grid data and site-level data. The large inter-model spread in CMIP6 $tas$ climatologies indicated substantial differences in model configuration or parameterization. The family resemblance in $tsl$ biases (e.g., CNRM, HadGEM/UKESM) points to structural similarities within model families. The reduced seasonal spread in JJA suggests that model performance was more consistent during snow-free periods, while snow dynamics played a role in winter divergence.

The excessive spread in JJA $tas$ and $tsl$ for specific CMIP6 models (CESM2, HadGEM3-GC31-LL, UKESM1.0-LL) points to seasonal over-responsiveness to summer variability. The excessive DJF $RS$ and generally positive $RB$ of $pr$ were shown in Group C, in contrast to the well-constrained $RB$ and $RS$ of $pr$ shown in Group L. However, both groups exhibited similarly large spreads and inconsistent biases in $snd$, suggesting that the snow-rain criterion remains a source of uncertainty for the frozen soil simulation. Each land-surface model's snow scheme—and its sensitivity to air and soil temperature—limits how much of that precipitation is accounted for as snow. The positive winter relative $pr$ bias and smaller, less systematic relative $snd$ bias across most models points to possible compensation mechanisms or misalignment between precipitation input and snow accumulation. With better $RS$ and $RB$ of $tas$ and $pr$ in Group L, models still had diverse abilities simulating $tsl$ and $snd$. This indicates challenges in accurately capturing the variability and relative bias for these specific variables across different models and seasons. The systematic over- and underestimations highlight the importance of further refining model approaches to improve performance in simulating snow and soil temperature dynamics. Snow in DJF, MAM, and SON (Fig. A1) and the

prevailing phase change processes in the soil were likely the main reason for higher $tsl$ interannual variability in these seasons. In JJA, all LS3MIP models showed low interannual variability, further indicating the role of snow in stable $tsl$ simulation.

     Given the smaller $tas$ $EB$ and the larger $tsl$ $EB$ in the LS3MIP than in the CMIP6 simulations, the bias spread introduced by the land-only models was large and is partially compensated for by the atmospheric component in the CMIP6 simulations. This explains why CMIP6 models show better $tsl$ results. As shown in the subplots of Fig. 5 ((C) tas-DJF) and (L) tsl-DJF),

atmospheric models in CMIP6 tended to show warm biases in air temperature ($tas$), while LS3MIP land models exhibited cold biases in soil temperature ($tsl$) at the same locations, particularly in Siberia. When driven by the same forcing, however, land models revealed their intrinsic tendencies, which also contributed to differences in snow accumulation despite similar precipitation inputs. The notable differences in $ES$ and $EB$ across seasons further emphasize the importance of understanding diverse land-surface interactions in different seasons. Moreover, the seasonal consistency of Group L's $tas$ $ES$ suggests that

the bias primarily arises from geographical features at station locations and how these were further represented at different model resolutions and surface configurations. In DJF, the multi-model $EB$ of CMIP6 and LS3MIP $tsl$ was negative, but it's positive for CMIP6 $tas$. And the $ES$ of LS3MIP $tsl$ was larger than $ES$ of CMIP6 $tas$, although LS3MIP models were forced by the same atmospheric data. This indicates larger variability caused by the land surface model than the atmospheric model.

### 4.3    Permafrost Region

The large RMSE and low correlations of $tsl$ in DJF, compared to better results in other seasons, highlight a seasonal dependency in simulation skill, possibly tied to limitations in frozen-process representations. The consistency of $tsl$ accuracy with $tas$ in JJA suggests that under unfrozen conditions, the models can simulate soil–atmosphere coupling more reliably. Moreover, better snow simulation ability improved soil temperature simulation performance. The lower $tas$ correlation in JJA than in MAM and SON indicates possible deficiencies in representing surface energy exchange during summer. The difference in the correlation

performance of SON $snd$ between Group L and Group C highlights the importance of a good simulation $pr$ during the snow accumulation period. Nevertheless, low correlations and high standard deviations of Group C $pr$ from spring to autumn suggest that improvements are still needed in simulating precipitation seasonality.

### 4.4    Climate Dependency of Modeled Temperatures

Cao et al. (2022) discussed an unreasonable warm $tsl$ bias that is possibly due to the overestimation of permafrost snow depth

in ERA5-Land. Similar warming of $tsl$ due to excessive $snd$ occurred in the LS3MIP simulation of HadGEM3-GC31-LL. Similar to findings from CLM5 simulations Dutch et al. (2022), several models (CESM2, CNRM-CM6.1, CNRM-ESM2.1, and MIROC6) exhibited cold $tsl$ biases despite warm $tas$, likely due to underestimated snow and soil insulation. This resulted in excessive energy loss from soil to atmosphere when $tas < -5\,°C$. There was an excessively low $tsl$ shown in Fig. 8. At 0.2 m depth, this cold bias mainly occurred when soil temperatures were below $0\,°C$. As shown in Fig. A4 and Fig. A5, at deeper soil

layers (0.8 m and 1.6 m), the cold bias was also present when soil temperatures were above $0\,°C$. This phenomenon exhibited strong model dependence, suggesting that the cause was not only related to insufficient representation of surface and snow

insulation but may also stem from factors such as overly strong thermal resistance in the land-only model parameterizations or low soil moisture content.

In addition, we categorized the model output by the freeze/thaw state of observation. This enabled us to compare the performance of two groups in different temperature states. To avoid assessment errors caused by different sample sizes, we discussed the overall uncertainty exhibited by the model in the thawed state, the freeze-thaw transition state, and the frozen state, using the boundaries of -5 °C and 5 °C, as the phase change process occurred most frequently between the boundaries. Low kurtosis of CMIP6 $tas$ difference shows that the CMIP6 ensemble fails to reproduce the realistic spatio-temporal variation of permafrost $tas$. The larger standard deviations in LS3MIP $tsl$ results compared to CMIP6 indicate that the land-only models produce higher variability in Set Intermediate and Set Frozen. The consistently negative $tsl$ bias implies that land models systematically simulate colder than observed soil temperatures. Notably, only in Set Intermediate LS3MIP and CMIP6 simulated high kurtosis. The close-to-zero median and high kurtosis in Group L $tas$ did not translate into similarly good results of $tsl$. This further demonstrates that land surface models exhibit distinct simulation tendencies in frozen soil. When driven by identical atmospheric forcing, the differences in LS3MIP $tsl$ among models are substantially greater than those observed in their corresponding CMIP6 simulations. Furthermore, the elevated IQR and standard deviation of the $tsl$ difference in Group L when $tsl$ was below –5 °C suggest that discrepancies in snow insulation representation, soil layering schemes, or freeze-thaw parameterizations contribute to greater simulation inconsistency within the model ensemble.

## 4.5 Snow Insulation

Two key phenomena should be considered regarding the snow insulation effect. Firstly, the insulating effect of the soil surface layer itself remained influential even when $snd$ is very thin. Secondly, the impact of snow on $\Delta T$ ($tsl-tas$) showed the greatest variability when the snow depth was shallow, stabilizing near 0 °C or lower once the snow depth exceeded about 0.2 m. This stabilization was minimally affected by $tas$, with inflection points reached at different depths depending on temperature ranges. The insulating effect of snow is due to its low thermal conductivity (about $0.3 \, \mathrm{W \, m^{-1} \, K^{-1}}$), which reduces heat loss from the soil. This effect became more pronounced at lower $tas$ as the temperature difference ($\Delta T$) increases.

The results from Fig. 10 suggest that factors beyond $tas$ and $snd$ impact the ability of the snow layer to impede energy transfer. Evaluating the snow insulation effect based on LS3MIP or CMIP6 runs alone may lead to different conclusions. For instance, the CESM2 LS3MIP run showed a snow insulation effect closer to observations, whereas the CESM2 CMIP6 run simulated a much lower $\Delta T$ curve. The issue with their snow insulation dynamics was the low initial insulation and slow increase in $\Delta T$ when $snd$ was below 0.2 m.

Four land models mentioned in this study are newer versions of the land models studied by Wang et al. (2016). Comparing Table A1 with findings of Wang et al. (2016) revealed notable advancements in newer models. CLM5.0 (indicated as CESM LS3MIP in Fig. 10) generated $\Delta T$ profiles that more closely resemble observed values within the range of -5 °C to -15 °C with an RMSE of 0.79 °C (Table A1) compared to CLM4.5 (RMSE of 1.46 °C in Wang et al. (2016)). However, in the colder categories, it did not perform better. The newer versions of JULES, used in HadGEM3-GC31-LL and UKESM1.0-LL, provided better initial insulation at low $snd$ values and did not overestimate the insulation effect at higher $snd$ values (lower RMSE in

all 3 categories). Compared to its previous version, ORCHIDEE (IPSL-CM6A-LR) exhibited a larger snow insulation effect that was closer to observed values (lower RMSE in all 3 categories). MIROC6 also showed lower RMSE compared to its older version in all three temperature categories. Furthermore, the study by Burke et al. (2020) also compared the insulation effect of various models, but only within the -15 °C to -25 °C category and among coupled climate models from CMIP6 and CMIP5,

not land-only models. Surprisingly, their results showed a degradation from CESM1 to CESM2 (CLM4.5 to CLM5.0) when it comes to representing snow insulation. Our results (CESM CMIP6 in Fig. 10) confirmed this finding, though the land-only simulations (CESM LS3MIP) performed better.

## 4.6   Impact of Land Model Features on Performance

The strong cold bias of CNRM models may be linked to how the models handle snow cover fraction. These models allow for

a snow-free fraction within vegetated areas, thereby reducing the thermal insulation effect and resulting in colder simulated soil temperatures. The weaker snow thermal conductivity in the CNRM models was also confirmed by the results in Fig. 10. As highlighted in Decharme et al. (2019) and Wang et al. (2016), this issue contrasts with observations, which are mostly free of intercepting vegetation. This exaggerates the snow's insulation effect in observations in the region. However, land surface temperatures in boreal forests are typically warmer in winter, mainly due to higher albedo compared to openly snow-covered

areas (Li et al., 2015). Regardless of differences in parameterizations or whether the simulations were coupled or uncoupled, all models analyzed underestimated soil temperatures in autumn.

Better snow and soil temperature (under snow) simulations are related not only to snow parameterization but also to other land model processes, such as soil boundary conditions. HadGEM3-GC31-LL and UKESM1.0-LL had good performance in simulating $tsl$ with their shallow soil column. Moreover, the zero-flux assumption is possibly more influential on the soil

surface when used in a shallower soil column; it constrains the $RS$ of soil temperature in winter. In general, defining a bottom flux showed better results than providing fixed values when considering the impact of bottom boundary conditions on soil temperature.

None of the models could accurately depict soil insulation. When $snd$ was below 0.05 m, all models simulated $\Delta T$ (Fig. 10) that was lower than the observed value. This shortage could lead to an overestimation of the total $\Delta T$ if the model soil surface

insulation is increased. The shortage could be due to insufficient representation of the thermal offset, which is controlled by factors such as soil texture, soil moisture, and surface organic matter. Organic layer can accumulate up to 15 cm on top of the surface of frozen soil with porosity greater than 0.95 (Boike et al., 2013). It is an important factor in the thermal dynamics of the soil surface (Zhu et al., 2019). Incorporating the impact of the surface organic layer improved the soil temperature simulation in land surface models (Ekici et al., 2014; Chadburn et al., 2015). It was especially true when some models adequately reproduce

the insulation magnitude at high $snd$. UKESM1.0-LL simulated large insulation effects of 4 to 12 °C under low $snd$ (lower than 0.05 m), which was closest to the observation and which might guide to further refinements. Soil moisture critically governs permafrost thermal behavior: high water content lowers frozen-soil thermal conductivity and heat capacity (Langer et al., 2011a; Jafarov et al., 2020). Low soil moisture content and the absence of an explicit phase change process can lead to a cold bias in model simulations, making soil temperatures more sensitive to atmospheric forcing. For example, models such as

IPSL-CM6A-LR exhibit rapid summer thawing, likely due to insufficient latent heat buffering from soil moisture (Burke et al., 2020).

The HadGEM3-GC31-LL, UKESM1.0-LL, and IPSL-CM6A-LR models exhibited the best soil surface insulation when considering the $RS$ and $RB$ of summer $tsl$. However, the significance of surface soil insulation was still underestimated.

It is challenging to identify how model features influence the vertical energy transportation process without conducting sensitivity experiments. Different physical processes represented in land surface models may interact in complex ways, either synergistically or oppositely, affecting the model's simulation capability. Without the ability to isolate the effects of these various processes, it is difficult to determine whether simulation errors resulted from one specific scheme or multiple overlapping processes.

However, it should be noted that the models with a better representation of snow insulation (Fig. 10), namely HadGEM3-GC31-LL, UKESM1.0-LL, and IPSL-CM6A-LR, use more recent formulations for snow thermal conductivity (Calonne et al., 2011; Wang et al., 2013). In particular, the updated snow thermal conductivity formulation in JULES (HadGEM and UKESM) from Yen (1981) to Calonne et al. (2011) improved the snow insulation effect, as demonstrated by a comparison of our results with Wang et al. (2013). MIROC6 was not the worst in terms of snow insulation, even though it assumes snow density and thermal conductivity to be constant. Furthermore, the low snow insulation in the CNRM simulations could not be attributed to its snow thermal conductivity formulation because it uses the same formulation as ERA5-Land. Their total insulation also depends on snow density and snow cover fraction.

## 5   Conclusions

This research investigated coupled CMIP6 and land-only LS3MIP historical climate simulations in frozen soil areas. Errors caused by the land surface models versus the errors caused by atmospheric forcing or coupled models were quantified and discussed.

Except in summer months, inaccurate interannual variability in the simulation of soil temperature by CMIP6 models is mainly caused by deficiencies in the land surface models and is less inherited from atmospheric components. Biases in the land surface models even partially compensate for the influence of air temperature biases. Similarly, improved precipitation simulation does not necessarily lead to better snow depth results in winter and spring. However, in autumn, better $snd$ was simulated with better $pr$ in LS3MIP. Land surface models performed better when coupled to an atmospheric model (CMIP6). Balancing tuning in coupled climate models with achieving physically accurate land-only simulations is a key consideration for future model development. Good soil temperature and snow performance in the coupled model do not necessarily indicate that the land surface component is responding realistically to atmospheric forcing.

Soil temperature biases and their spread between models are much more evident in winter than in the other seasons. Spatially, the models exhibit larger disagreements in reproducing the soil temperature of permafrost sites. The largest bias standard deviation disagreements of $tas$ and $tsl$ are observed in Set Frozen (temperature lower than -5 °C) for both ensembles (see Table 3). These indicate a limitation of models reproducing the $tsl$ relationship with $tas$ in freezing conditions. Land surface

models need to incorporate or improve processes related to frozen soil and soil hydrothermal dynamics in frozen conditions. This includes enhancing the simulation of soil moisture content, refining soil thermal and hydraulic parameterizations in frozen

states, and representing key features of frozen soil, such as excess ground ice and surface organic matter.

The deficiency of land surface models is reflected in the ability to simulate snow depth and/or to represent the effect of thermal insulation. Snow insulation plays a critical role in modulating soil temperatures. Updating the parameterization of snow thermal conductivity, as demonstrated in recent models such as HadGEM3-GC31-LL and UKESM1.0-LL, can enhance the insulation effect of snow. However, the insulation effect is not solely determined by thermal conductivity parameterization.

Accurate parameterizations of snow density, snow depth, and snow cover fraction are also important factors. In particular, snow cover controls the spatial continuity of insulation. Even when snow depth is accurately simulated, inadequate thermal insulation can result from insufficient snow cover because exposed ground patches allow greater heat loss. Addressing these factors in future model development is essential for improving the representation of snow insulation and, consequently, soil temperatures.

Note that the main scope of this study is limited to soil depths down to 0.2 m and that the thermal state of frozen soils is not determined solely by temperature (Groenke et al., 2023). It is essential to consider various hydrothermal processes within the deeper soil, including thermal offset, permafrost active layers, seasonally frozen soil freezing depth, and heat and water transport. These factors play a critical role in capturing frozen soil dynamics and must be investigated further.

*Author contributions.* ZL and BA determined the research outline and methodology, and wrote the initial manuscript. ZL and DR analysed
the results and edited the manuscript. BA provided guidance on data analysis. ZL collected the data, generated the figures, and was responsible for the code calculations. Results were discussed by all authors.

*Data availability.* CMIP6 and LS3MIP multi-model ensemble data (http://doi.org/10.22033/ESGF/CMIP6.4066, Voldoire (2018), http://doi.org/10.22033/ESGF/CMIP6.4095, Voldoire (2019b), http://doi.org/10.22033/ESGF/CMIP6.4095, Seferian (2018), http://doi.org/10.22033/ESGF/CMIP6.9599, Voldoire (2019a), http://doi.org/10.22033/ESGF/CMIP6.5195, Boucher et al. (2018),
http://doi.org/10.22033/ESGF/CMIP6.5205, Boucher et al. (2019), http://doi.org/10.22033/ESGF/CMIP6.5603, Tatebe and Watanabe (2018), http://doi.org/10.22033/ESGF/CMIP6.5622, Onuma and Kim (2020), http://doi.org/10.22033/ESGF/CMIP6.6109, Ridley et al. (2019), http://doi.org/10.22033/ESGF/CMIP6.14460, Wiltshire et al. (2020b), http://doi.org/10.22033/ESGF/CMIP6.6113, Tang et al. (2019), http://doi.org/10.22033/ESGF/CMIP6.14462, Wiltshire et al. (2020a), http://doi.org/10.22033/ESGF/CMIP6.7627, Danabasoglu (2019b), http://doi.org/10.22033/ESGF/CMIP6.7650, Danabasoglu (2019a)) were downloaded from https://esgf-node.llnl.gov/projects/cmip6/ on 22-
05-2023.

ERA5-Land monthly averaged data from 1950 to present. Copernicus Climate Change Service (C3S) Climate Data Store (CDS) http://doi.org/10.24381/cds were accessed on 08-08-2024.

The daily observational data from RIHMI-WDC can be collected from http://aisori-m.meteo.ru/waisori/

*Competing interests.* The contact author has declared that none of the authors has any competing interests

*Acknowledgements.* Zhicheng Luo gratefully acknowledges the China Scholarship Council (CSC) sponsorship for Z.L. (No.202006040064). BA acknowledges support by DWD IDEA S4S – project FS-SF (4823IDEAP2). This work used resources of the Deutsches Klimarechenzentrum (DKRZ) granted by its Scientific Steering Committee (WLA) under project ID bb1064 and of Goethe-HLR. The authors thank Professor Duoying Ji from Beijing Normal University for providing valuable insights and support for this research. Thanks to Mittal Parmar for her contribution to this research. We express our gratitude to the World Climate Research Programme for its coordination and support of CMIP6 through the efforts of its Working Group on Coupled Modelling. We appreciate the climate modeling groups for producing and providing their model output, the Earth System Grid Federation (ESGF) for archiving the data and ensuring access, and the multiple funding agencies that support CMIP6 and ESGF.

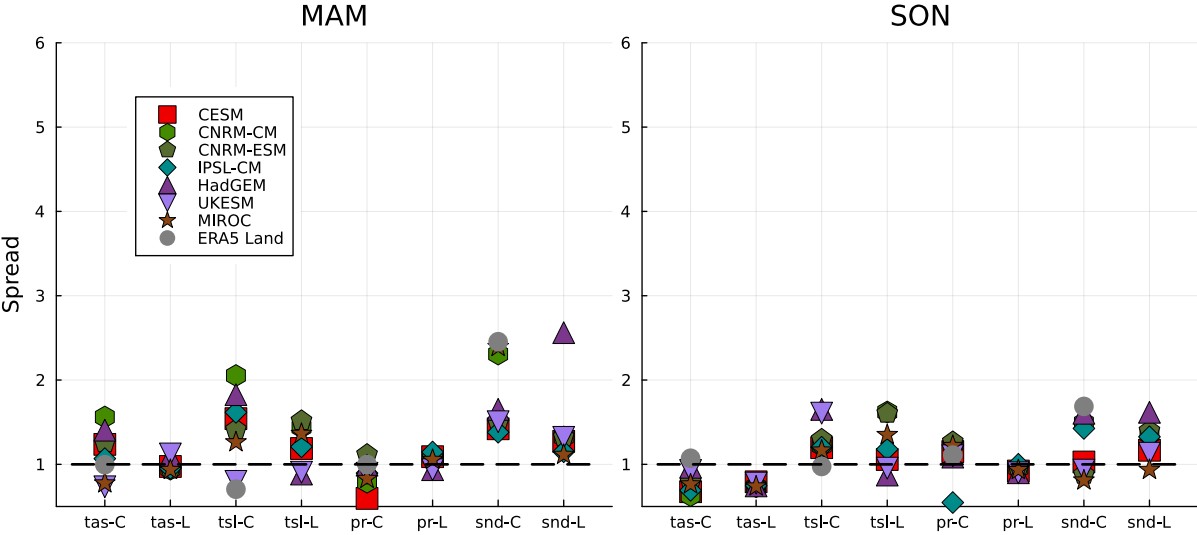

**Figure A1.** Relative spread ($RS$) of the sites-averaged climates (1985–2014) of the four variables in both ensembles and in E5LC with reference observations for all seasons. The colors indicate the models, and the x-axes show the variables and ensembles (C indicates CMIP6, and L indicates LS3MIP runs).

## Appendix A: Additional Figures

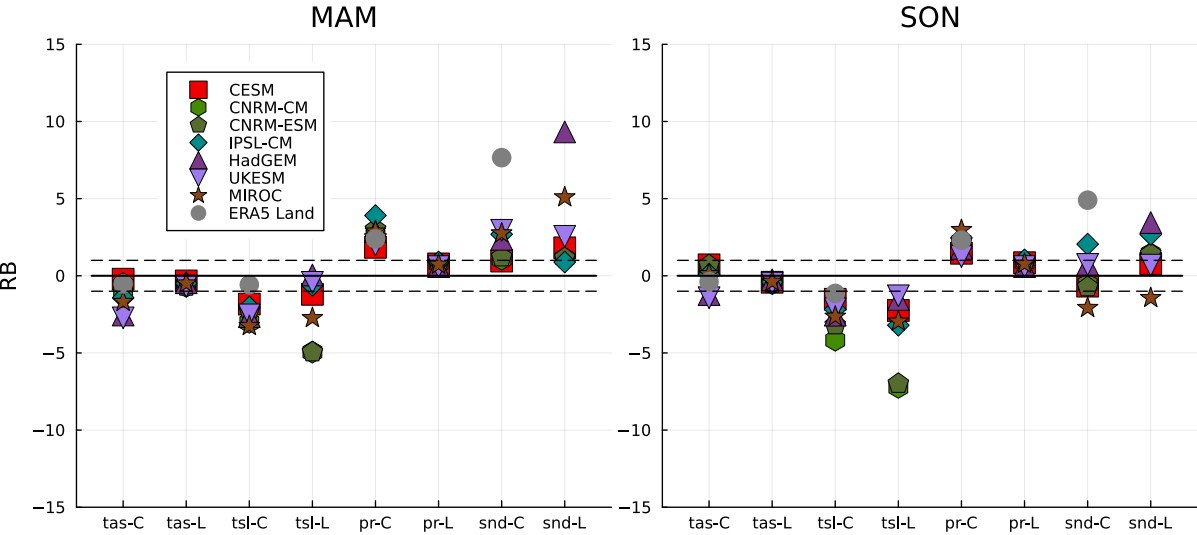

**Figure A2.** Same as in Fig. A1, but for relative biases (RB). The dashed lines (from -1 to 1) indicate the range of absolute median differences smaller than the observation's IQR.

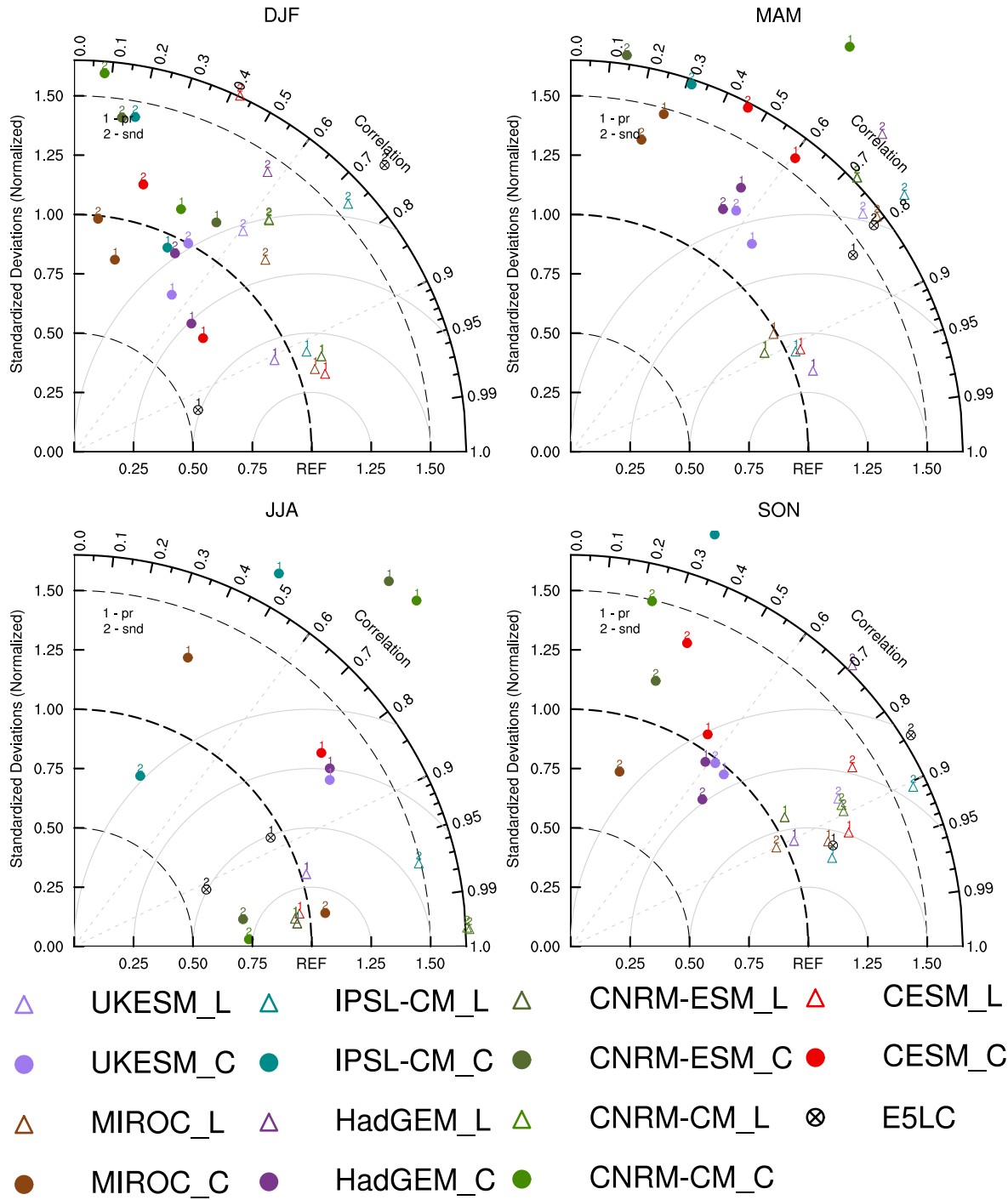

**Figure A3.** Taylor diagrams indicating the spatial correlation, standard deviation, and unbiased root-mean-square (RMS) difference (gray circles) for the four seasons of the simulated variables $pr$, $snd$ (labeled by numbers) against observations at permafrost sites (here: sites with mean winter $tas < -25\,°C$). Normalization is applied using the standard deviations of the observations. Colors indicate different models and black crossed circles E5LC. Triangles show the results of the LS3MIP runs, and solid circles of the CMIP6 runs.

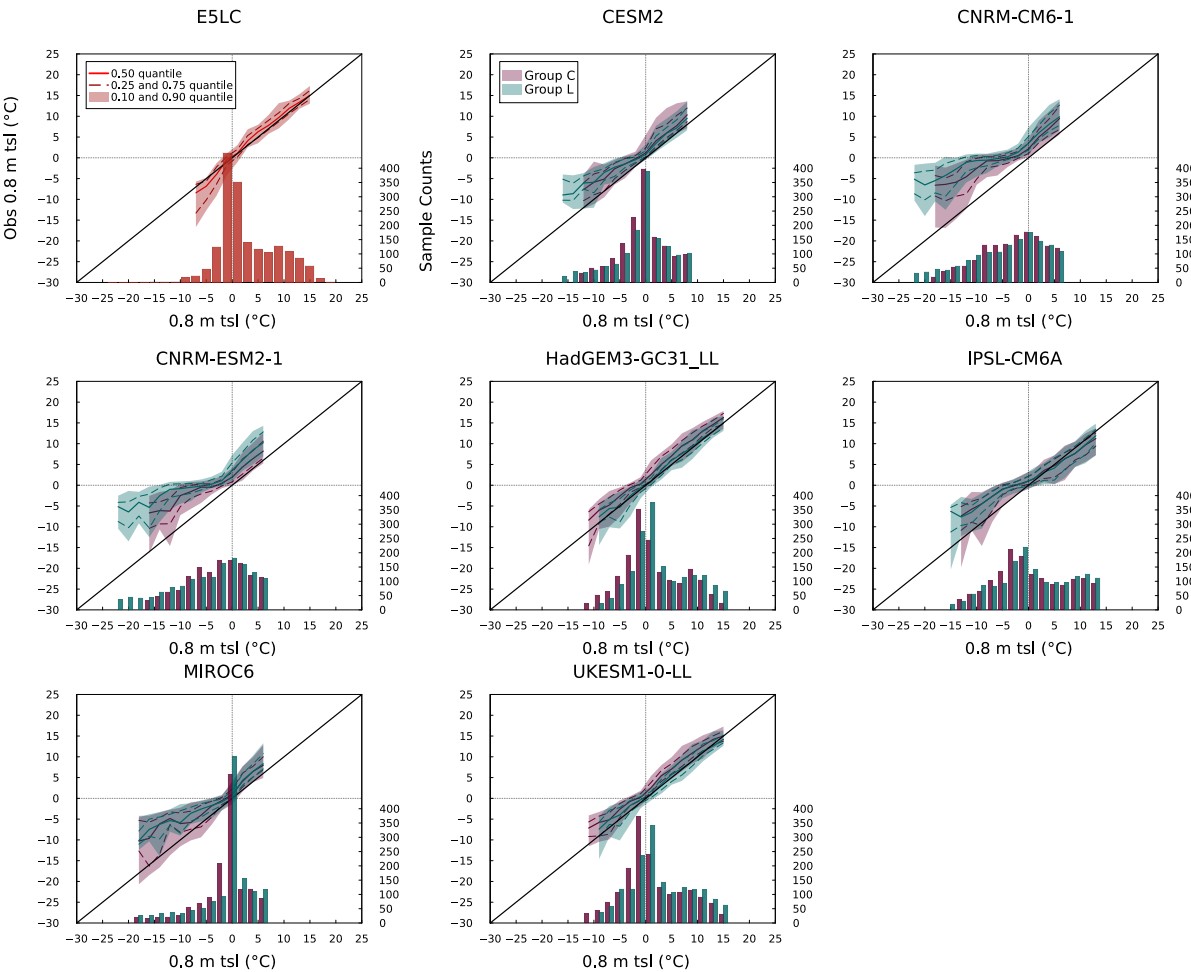

**Figure A4.** Quantile-Quantile (QQ) and histogram plots for soil temperature in depth 0.8 m. The plots show 30-year mean monthly 0.8 m $tsl$ data of models at all sites and their observational values. The model simulation outputs are binned at 2 °C intervals. The colored solid and dash curves are the median and 1st/3rd quartiles of the corresponding observations for all data points in the temperature interval, and the shaded area is the inter-decide range. The histograms represent the sample size within each temperature interval, and temperature intervals with sample sizes smaller than 20 are excluded from the Q-Q plots.

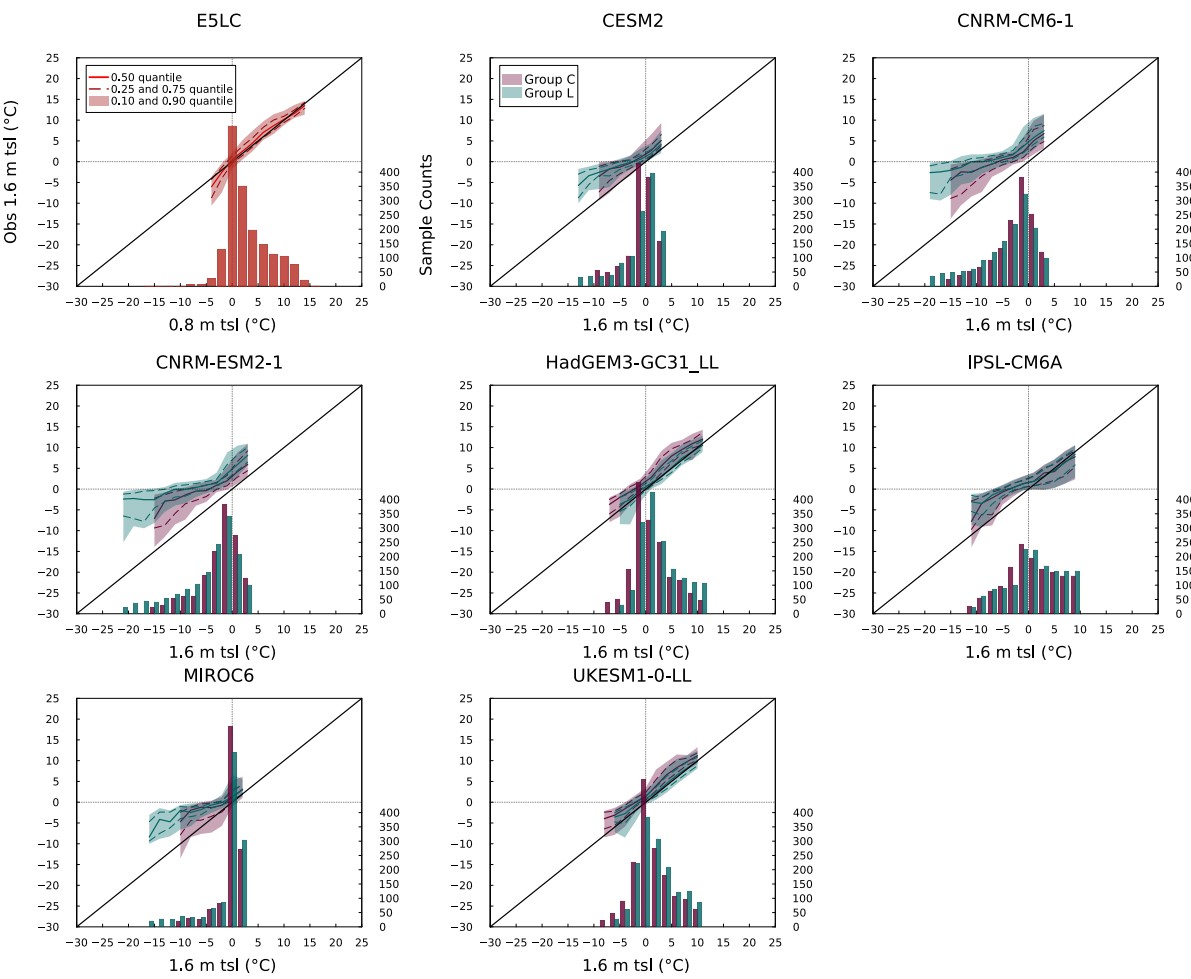

**Figure A5.** Same method as Fig. A4, but for soil temperature in depth 1.6 m.

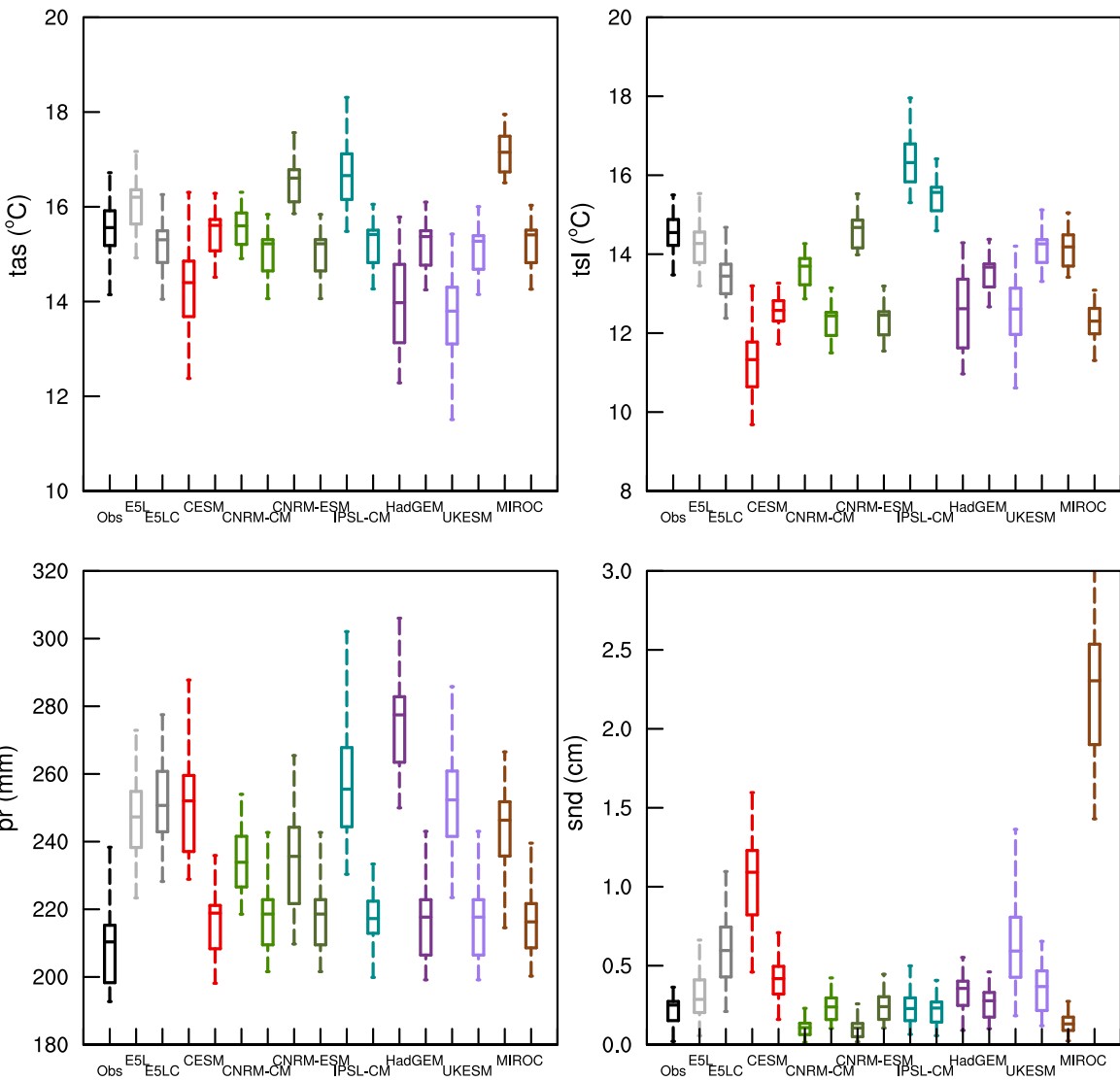

**Figure A6.** Sites' locations averaged JJA-climates (1985–2014) of hydrothermal variables as observed and simulated. Names on the x-axis and the colors indicate the different data sources. Model names indicate CMIP6 model output (left), and LS3MIP (right; see Table 1). Each CMIP6-LS3MIP pair shares the same color. The boxes represent the medians, first and third quartiles; the ± 1.5×IQR or the maximum and minimum values, if within the former range, are taken as the whiskers' length.

**Table A1.** Root-mean-square error (RMSE) between modelled and observed snow insulation effect (relationship between snow depth and soil-air temperature differences) across three $tas$ categories, within the 0–0.8 m snow depth range (excluding fill-values) in °C, as illustrated in Fig. 10.

| Model | -15 °C to -5 °C | -25 °C to -15 °C | less than -25 °C |
|---|---|---|---|
| E5LC | 1.05 | 1.52 | 2.73 |
| CESM (CMIP6) | 2.09 | 3.97 | 6.22 |
| CESM (LS3MIP) | **0.79** | 2.18 | 4.09 |
| CNRM-CM (CMIP6) | 4.83 | 9.58 | 12.39 |
| CNRM-CM (LS3MIP) | 6.30 | 11.47 | 16.01 |
| CNRM-ESM (CMIP6) | 4.75 | 9.67 | 12.15 |
| CNRM-ESM (LS3MIP) | 6.20 | 11.41 | 15.75 |
| IPSL-CM (CMIP6) | 2.26 | 2.79 | 4.34 |
| IPSL-CM (LS3MIP) | 1.56 | 3.25 | 3.94 |
| HadGEM (CMIP6) | 1.09 | **1.05** | **2.33** |
| HadGEM (LS3MIP) | 1.26 | 1.43 | **1.57** |
| UKESM (CMIP6) | **0.71** | **1.12** | 2.56 |
| UKESM (LS3MIP) | 1.20 | **1.16** | **2.15** |
| MIROC (CMIP6) | 1.11 | 2.14 | 5.41 |
| MIROC (LS3MIP) | **0.79** | 1.72 | 4.32 |

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
