# Peer review of "Assessing Uncertainties in Modeling the Climate of the Siberian Frozen Soils by Contrasting CMIP6 and LS3MIP"

_EGUsphere, 2025_

## Author Response (AR1)

We would like to thank the reviewers for their detailed and comprehensive comments and suggestions. We greatly appreciate the time and effort you have devoted to reviewing our manuscript. Below we respond to each comment point by point. The reviewers' original comments are in **black**, and our responses are in **blue**.

**For RC1:**

**General comments:**

ERA5-Land: The presentation of the ERA5 data in the manuscript seems pointless. The purpose of the manuscript is not to evaluate ERA5 Land against station data. ERA5 Land is also not used for additional validation of the model simulations on a larger spatial scale than the station observations allow. I suggest to remove the ERA5 Land contributions in the manuscript, or actually make use of them within their limitations (which would then warrant their evaluation against the station data).

We have revised our approach to the presentation of ERA5-Land data. We no longer use ERA5-Land as a "benchmark" or reference dataset for model evaluation. Instead, ERA5-Land is now treated as a widely used dataset that serves as a complementary basis for comparison. Similar to the LS3MIP simulations, ERA5-Land is a land-only model driven by reanalysis. In addition, the ERA5-Land data have been regridded to match the resolution of the selected LS3MIP simulations (~0.9°). This adjustment allows us to assess how differences in resolution affect the results. We will also expand the Data and Methods section to provide further context for these changes.

Discussion: The manuscript lacks a proper discussion of its results in a comprehensive way, which also leads to conclusions that seem to have no basis. In the introduction, you state that: (1) "We will analyze the discrepancies between the same model in CMIP6 and LS3MIP to quantify the bias and uncertainty present in frozen soil regions, attributing them to land surface models versus those resulting from atmospheric forcings. With identical and more realistic atmospheric conditions, we anticipate that the LS3MIP models will more accurately simulate soil conditions. If these models fail to produce soil variable outputs that align better with observed data than the CMIP6 simulations, it is regarded as an error in the land surface models." (2) "We will discuss the variations among different models of LS3MIP and try to establish a connection between model performance and their specific features." However, the manuscript ends after the presentation of the results, without coming back to the analysis you promise in a comprehensive way. Since the you do not have a discussion section in the manuscript, the conclusions need to contain this discussion (or you need to make a discussion section). Please come back to both points from the introduction, and establish conclusions for both points rooted in your results in an understandable way. Right now, some conclusions and discussion are scattered throughout the results, but it is hard to puzzle them together to a coherent picture.

Indeed, we recognize that the current structure of the manuscript and the content within the conclusions do not sufficiently address these two important points. To enhance

clarity, we will separate the results and discussions. We will include discussions that directly respond to both points outlined in the introduction. The additional sections will focus on "Discrepancies Between CMIP6 and LS3MIP" and "Differences in LS3MIP Model Performance".

**Specific comments:**

Thank you for your valuable and specific comments, which we greatly appreciate. We will take them into account in our revisions to improve the manuscript. In particular, the introduction will be rewritten and the results and discussion sections will be separated. Below, we address some of your comments individually where further clarification or alternative perspectives are needed; for other comments, we agree with your suggestions and will revise accordingly.

**1 Introduction**

The introduction loosely strings together statements on Arctic climate change and its impacts on the Siberian permafrost region, then jumps to factors that determine permafrost thermal state, and finally barely introduces CMIP6 and LS3MIP. Without knowing all of these things already, and how they are related to the uncertainties eg in the permafrost carbon feedback from climate model projections of the future, it does not tell the reader much, and does not coherently argue the importance of this study. Please outline a clear relation between the facts mentioned in the introduction and the statements about what the paper means to do from the last two paragraphs. I suggest rewriting the introduction completely. In addition, I have a number of specific comments on the introduction below.

We will adjust the structure of the introduction and add content to clearly explain the role of permafrost in climate change, the role of climate models in permafrost research, and the challenges of permafrost modeling. This will provide a clearer logical chain explaining why we are conducting this research.

Line 20: This is rather vague, please give specific numbers to the magnitude of Arctic Amplification, and cite their sources.

The data presented originated from calculations of temperature increase rates derived from the CMIP6 model ensemble across different regions; however, these contents were not specifically detailed in the manuscript. Moreover, our claim of "twice" referred specifically to areas within the Arctic permafrost zone, not to the broader Arctic region. To clarify and provide accurate information in accordance with your feedback, we will revise this statement.

Line 20: While I don't doubt the numbers for Arctic climate change cited from the two papers, and I acknowledge that they are permafrost related publications, these aren't the papers that produced the numbers, and they are seriously outdated. Please cite more recent publications on climate change projections for the Arctic, and cite the direct

sources.

Climate simulations indicate that the Arctic warmed at a rate of 0.66 ± 0.32 °C/decade from 1979 to 2014 (Cai et al., 2021), and high-latitude warming in a moderate scenario of 1.2 to 5.3 °C from 2005 to 2100 (Koven et al., 2013).

Line 22: While it is certainly true that the most distinct impacts occur in the permafrost areas where temperatures are already close to zero, the statement seemingly has no relation to your manuscript, since you focus your detailed analysis on cold regions, not the warmer edge of the permafrost zone, so I don't see the relevance of that statement.

While it is true that our primary focus is in permafrost regions, many of our selected stations are situated at the warmer boundary of permafrost degradation. Furthermore, in Section 3.4 (Figure 9), we examine the implications of reaching the critical temperature threshold of 0°C on model simulation accuracy. Our aim is to investigate whether models may exhibit greater simulation errors under climate conditions characterized by significant permafrost degradation. Therefore, although our main analysis centers on cold regions, understanding these warmer boundaries is crucial for assessing overall model performance.

Line 27: Again, this is very vague and lacks an appropriate reference. Please clarify.

Large amounts of soil carbon are stored in permafrost (Tarnocai et al., 2009; Fuchs et al., 2018), and as permafrost thaws, the soil carbon may be released into the atmosphere at a faster rate (Schädel et al., 2018).

Line 28: The point about abrupt thaw is that from models. we can usually only estimate the carbon emission effects of gradual thaw, but the effects of abrupt thaw are expected to be substantially larger than those of gradual thaw. However, instead of saying that, you simply line up facts with no connection or argument. Please rephrase, and cite appropriate sources.

In environments such as lakes and wetlands, the impact of thawed carbon on climate is even more pronounced due to the low-oxygen conditions, further increasing the proportion of methane emitted along with other greenhouse gases (Koven et al., 2015; Walter Anthony et al., 2018). Processes such as thermokarst result in sudden thaw events that greatly enhance the decomposition and release of frozen soil carbon, potentially increasing carbon emissions by up to 50% (Abbott and Jones, 2015; Turetsky et al., 2019).

Line 33: There needs to be at least one general, bridging sentence on how heat transfer through the soil is simulated, and that the following paragraph speaks about modelling.

In land surface models, heat transfer through the soil is typically simulated as one-dimensional vertical transport. Models adopt their specific soil layering schemes, where the thickness of soil layers generally increases with depth. By calculating the water and thermal balance at different depths, land surface models can derive the current state of soil moisture and temperature.

Line 35: "There are differences in the time scales of major physical processes between the soil and the atmosphere." Vague, please clarify what you mean.

The timescales of important processes differ largely between the soil and the atmosphere. For example, key variables such as temperature and humidity in the near-surface atmosphere can fluctuate substantially over hours or even minutes. In contrast, changes in water and thermal states in the soil are much slower with depth; for example, at depths of tens of meters below permafrost, soil temperatures may not vary significantly for decades.

Lines 33-42: These two paragraphs are a weird mix of processes and conditions controlling heat transfer through the soil, and how these are represented in models. Please separate clearly.

We will rewrite.

Line 43: Please state what CMIP6 means.

Information added.

Line 46: There are a number of papers that describe these advances that should be cited here.

We include here the following references:  (Ekici et al., 2014; Chadburn et al., 2015; Decharme et al., 2016; Brunke et al., 2016; Jafarov and Schaefer, 2016; Guimberteau et al., 2018; Cuntz and Haverd, 2018; Damseaux et al., 2025).

Line 47: Please state what LS3MIP means. Also, introduce what LS3MIP aims to do before you dive into the protocol.

Information added.

2.1 CMIP6 and LS3MIP Simulations

Line 75: Land models treat input data differently, and may require different forcing data sets per se. A table would be nice, in particular since you look at tas, which can be close to/identical to the forcing, or quite different, depending on model setup.

We will double check the GSWP3 data that used in LS3MIP, if the models show strong differences we will add more information in the results (e.g. in Figure 2).

Line 89: Ménard et al only show snow properties in their paper. The way you cite the paper implies all information in you table can be found there, which is not the case. Please clarify.

We rephrased this line and supplemented our table about snow properties with further literature.

Line 89-90: This is very vague again, and the table misses some of the processes mentioned here. Eg how is vegetation represented, are the Arctic specific vegetation types, are there shrubs? Please clarify and expand your table.

We will add necessary information to Table 2, such as the vegetation type options/distribution, as it may affect snow accumulation, albedo, and surface soil insulation.

Line 96: I find this sentence misleading, it implies that models that consider the impact of surface organic matter with a focus on hydro-thermodynamics don't include a carbon cycle, which is for example wrong for CLM5. Please rephrase.

We removed this part as we are not discussion about organic matter here.

Table 2: Power Function and Quadratic Equation: What does that mean? Either explain somewhere, or use a more descriptive term. What does snow conductivity depend on in these equations?

The terms "power function" and "quadratic equation" refer to the mathematical formulations used to describe the relationship between snow thermal conductivity and snow density (see Menard et al., 2021 and Wang et al., 2016). However, to improve clarity and address your concerns, we have replaced this ambiguous phrasing with explicit references to the specific formulations.

**3 Results and Discussions**

3.1 Winter 2-m Temperature in Target Area

Line 152: The definition used in Lawrence and Slater is the generally accepted definition of permafrost. Quite a number of the stations denoted as circles are actually situated on permafrost. Please explain potential reasons why they are not categorized as permafrost using this definition on the station data.

The definition of permafrost, which requires at least two consecutive years of soil layers in a frozen state, poses two problems when implemented in this study. 1. Many sites lack deep soil layers below 0.8 m, and the soil above 0.8 m at these sites almost always thaws annually. 2. The study spans 30 years, during which some sites had frozen layers for two consecutive years, but the most of the time was characterized by freeze-thaw cycles. We have therefore adopted a very strict permafrost screening scheme: a site is defined as permafrost only if the observation data at a depth of 3.2 meters (with sufficient observations) remain completely frozen throughout the 30-year period. Of course, it is not rigorous to define only these sites as permafrost, we will come back to the method of Lawrence and Slater.

Figure 1: The two triangle stations are very hard to see. In general, the figure would convey more information if the stations were colored by bias in comparison to the modelled data instead of their own mean states. Also, It would be useful to show the

permafrost boundaries either from Brown et al or Obu et al in the map.

Thank you for your feedback regarding the visibility of the triangle stations. We have increased the size of the triangles to improve their visibility. Additionally, we will use the modeled temperature as a base map and represent the differences between station data and model outputs with color coding for clarity. So far we find that the model ensemble will have some grids being recognized as sea as the resolution is much lower than ERA5 Land. We cannot have the bias between model ensemble and observation of some sites due to this (especially triangles sites). So we will remove those sites from Figure 1. We appreciate your suggestion to include permafrost boundaries from either Brown et al. or Obu et al.; however, adding this could clutter the figure, so we do not plan to include it.

Figure 2: Bars need to be broader, median positions are not visible. Also, the labels have no positions, which makes them meaningless. For precipitation and tas, we could learn a lot from seeing where GSWP3 is, since it is the forcing data.

Regarding the readability of the figure, we have made adjustments to ensure that the bars are wider and that median positions are clearly visible. And I have altered the positioning of labels.
Regarding GSWP3, it is important to note that all LS3MIP models' temperature and precipitation results essentially derive from GSWP3 at varying grid resolutions. Therefore, we believe it is unnecessary to include an additional box for GSWP3, as this information can still be referenced without it.

3.2 Model Climatologies

Line 160: Looking at figure 2, I don't see that.

As we improved the readability of the figure it should be more clear now.

Line 162: This statement is only true for pr. The LSMs compute their own tas. How close that actually is to the forcing depends a lot on what forcing is used (eg temperature at a reference height, or 2m air temperature itself), and on how complex the calculation within the LSM is.

We revised as :"*Even though the LS3MIP models were forced with identical reanalysis data, their tas and pr medians differed slightly, as they have different grid resolutions and methods for calculating and regridding tas and pr.*"

Line 169: What about snow, soil moisture, vegetation? There is a distinct difference between soil temperatures in general and TTOP, which refers to (1) mean annual temperatures and (2) the top of the permafrost table.

We removed this sentence from discussion.

Line 171: What does model family mean? Is it based on similarity of the atmospheres, or based on the atmosphere and land components? In your example, both land and

atmosphere components of the models you put into one family actually share code history, but since you do not even state if you refer to the LS3MIP or the CMIP6 simulations, the statement is unclear.

*We refer to Kuma et al. (2023) about the definition of model family. And here we define it by their land components. We will make a clearer statement about this.*

Line 179: What is the reason for this difference in snow? Precipitation is similar, at least for winter, and air temperatures differ, but are so far below zero that the difference seems irrelevant. What drives this? Precipitation and temperature in autumn? And why does it only occur for this one model?

*We checked the HadGEM-LS3MIP data and found that in other seasons, the tas and pr values of HadGEM-LS3MIP were not significantly different from those of other LS3MIP models (especially UKESM). Furthermore, in MAM and SON, the air temperature and soil temperature of HadGEM-LS3MIP were at higher levels in the model ensemble, but snd still showed the same high values as in DJF.*
*We also compared HadGEM-LS3MIP and UKESM-LS3MIP on a site-by-site basis, and the phenomenon of overestimation of snd was widely observed at most sites. We have not yet found a satisfactory explanation for this phenomenon. We believe that the phenomenon is likely related to the snow scheme of this model. We will upload our scripts on Zenodo for transparency and reproducibility.*

Line 180: ±10 cm translates into a relative error of around 33%, which is massive! Please put into context.

*Revised as : "The models are diverse in simulating DJF medians $snd$, with high temporal variability."*

3.2.1 Relative Spread and Relative Bias

Figure 3: Caption states you show all seasons, yet there is only winter and summer in the figure. Also, why is snow in winter similar between L and C even though precipitation differs considerably? Because the medians are the same, and that is what drives snow variability? Please expand.

*We plotted the other seasons, but since we did not draw solid conclusion from these two seasons we decided to put them into appendix if necessary.*
*Figure 3 illustrates the relative spread of data over a 30-year period, representing the degree of variability. While precipitation is an important driver of snow accumulation, snow variability is also influenced by other factors such as air temperature and soil temperature. As that not all stations are located in permafrost regions. Therefore, higher temperatures and soil temperatures in certain models may limit the proportion of precipitation that accumulates as snow while they have higher relative spread of precipitation.*
*The ability of different land surface models to accumulate snow is constrained by their*

respective parameterization snow schemes. As shown in Figure 2, inter-annual variability for snow depth tends to be much lower than that for precipitation. C and L models derive their precipitation from different sources; thus their relative spreads naturally differ largely.

Figure 4: There is no shading. Correct the caption. Also, as Figure 3, this is not showing all seasons.

Fixed, the other seasons are added in appendix.

Line 186: In general, it is really hard to understand the summer parts of Figures 3 and 4 without an equivalent to figure 2. Maybe provide a summer version of figure 2 in the supplement. Specifically, I think this is meant to read "contrary to JJA where" or something similar. The sentence does not make sense as it is.

We revised this sentence to make it straightforward. The JJA version of figure 2 is added in appendix.

Line 193: "The pr in Group C exhibits more extensive group diversity than in Group L." Which is because in group L, the only difference between the different models is different interpolation of the forcing data set, which makes this statement meaningless.

We will delete this sentence.

Line 198: "the model's bias is considered relatively small" I would suggest to rephrase that into something like "the model's performance is considered adequate", because if the IQR is big enough, very big relative biases could still lead to RBs around 1. In terms of model performance, because you only look at 30 years of data, I agree that this means model performance is adequate, however, the bias would not be small.

We agree and will change the expression.

Line 200: "Almost all CMIP6 and LS3MIP models have a positive pr-bias but a smaller relative and non-systematic snd-bias in winter." Since snow is not a pure winter phenomenon and snow build up starts in autumn, so I am not sure how much meaning this comparison has. This analysis needs to be extended to snow build up in autumn.

We will expand this analysis to whole snow accumulation period.

3.2.2 Spatial Heterogeneity

Figure 5: I find this figure extremely irritating. Figure out the orientation, and resort so that maybe there are eight rows and two columns, so that the figure can be read. Also, for tas, the spread in the CMIP ensemble is bigger than the spread in the LS3MIP ensemble. For tsl, it is the other way around. Why? Please expand in the manuscript text.

We have added clearer info to this figure. After consideration we decide not to split the

figure as to better compare the difference between tas and tsl.
There is an explanation for this question in 218. It is due to the compensation phenomenon of climate models. In general, the land models have the opposite bias to their atmospheric models. And so they have more neutral and realistic results. Otherwise, if the atmospheric model has a warm bias and the land model also has a warm bias, it will produce unrealistically warm land. But if the models have the same forcing, the land models will show their natural bias.
Besides, modeled soil temperatures are more diverse in absolute value/average state (as in Figure 2). As ensemble spread counts for absolute values, tsl should has larger sizes than tas.

Line 218: Why would there be a compensating effect like that? The ensemble spread is not particularly strong in your figure. Please explain.

You can refer to our last reply. Please focus on the subplots in the upper left and upper right of Figure 5, which represent the tas simulated by the atmospheric model in CMIP6 and the tsl simulated by the LS3MIP land surface model (where the biggest circles are more than 10 °C standard deviation), respectively. In the model ensemble, the two have opposite biases at most sites in Siberia. In addition, you can observe the bias tendencies of each model in Figures 7 and 8, where this phenomenon can also be observed.

**3.3 Permafrost Region**

Figure 6: It is impossible to read the labels. If all variables are to be presented in one Taylor diagram, they need to be distinguishable.

We will separate the figure into two, one with tas and tsl, another with pr and snd (which will be put into appendix). Thus it can be more readable.

**3.4 Climate Dependency of Modeled Temperatures**

Line 245: In the figure caption, it says 50th quantile, eg median, instead of the mean, which actually makes more sense. Please check.

This is an oversight. For statistical reasons, we have used the median rather than the mean in this study.

Line 268: I think this needs to read "Four models …"

We revised the sentence to avoid confusion. Now as :"…*several models (CESM2, CNRM-CM6.1, CNRM-ESM2.1, and MIROC6) exhibited cold tsl biases despite warm tas, likely due to underestimated snow and soil insulation*."

Line 270: The reference is misleading, Dutch et al 2022 only discuss simulations with CLM. Please correct.

We will clarify this as: "*one conceivable explanation is that they underestimate the snow insulation effect (in line with the CLM5 simulations performed by \cite{dutch_impact_2022})*"

Line 272: "There is an excessively low tsl shown in Fig.8, possibly due to insufficient geothermal (functions as upward energy flux from the bottom of soil columns). As the decrease in tas has a limited influence on the tsl through high snd, the main source of error is likely from the other side of energy transportation (thermal conditions in the bottom of the soil column)." If that was true, models that consider a non-zero flux condition at the lower boundary would have to perform better than those with zero flux conditions, which is not the case. The depth of the column plays an important role here, as eg discussed in Alexeev et al, 2007 https://doi.org/10.1029/2007GL029536 and more recently Hermoso de Mendoza et al (2020), https://doi.org/10.5194/gmd-13-1663-2020.

Yes, the low tsl may be due to other factors. We will carefully analyze it and provide a more reasonable explanation.

Line 275/276: What about the strong underestimation of variability in summer? What is the reason for that?

We didn't add information about the variability in JJA here. However, in Figure 4 the LS3MIP underestimate summer tsl variability, and tas having almost identical spread with Obs. One reason could be without snow in summer, there is less uncertainty in tsl simulation. The models have less uncertainty considering impacts of soil conductivity and vegetation.

Line 285: "In contrast, ..." I don't understand that sentence. Please reformulate.

What I mean is that in LS3MIP's Set Intermediate and Set Warm, the standard deviation of the tas difference between model and obs is relatively small compared to CMIP6. And there is just a slight cold bias.

Line 292: That is a really important statement, it should be explicitly taken up in the conclusion, and the implications should be discussed!

We have mentioned it in the conclusion. We will expand the content about this in discussion and conclusion.

3.5 Snow Insulation

Figure 10: It would be really useful to have horizontal grid lines (maybe in light grey) in the figures so the reader can better understand how close to the observed values models are.

The horizontal lines are added.

Figure 10: CESM: This actually looks a lot better than what is Burke et al, 2020, for just winter. I wonder why.

Our assumption is that the CESM2 in the CMIP6 historical run has a very strong warm bias in tas. The CMIP6 results are much worse than LS3MIP for CESM2, and for the colder 2 categories, CESM2 doesn't have enough high snow depth samples.

If we focus only on the CMIP6 results of CESM2 and add up 3 categories, it should look like the result in Burke et al, 2020. As it goes from 5 to 10 °C in the -25 to -15 category.

Line 297: From your figure caption, I assume that you use monthly mean values from all months, not just the winter months, for your plot. However, I assume the classification is still based on the DJF 30 year average of the station?

No. The obs sample size is 12months*30years*236stations. The classification is based on the air temperature value of every sample not on the DJF 30 year average of the station. We will clarify it the caption and in the text.

Line 303: "under sufficiently thick snow, the tsl gradually convergences near 0 °C and is primarily impacted by tas in a limited manner." I don't understand that statement. Please reformulate.

We will reformulate. What we meant here is that tsl has an almost constant value (0°C or lower) if there is about 0.2m or more snow depth.

Line 325: "UKESM1.0-LL consistently demonstrated similar snow insulation effects in both ensembles" From just looking at the figure, so does HadGEM, which is not surprising since the land models are similar. MIROC and IPSL also have very similar curves regardless of the forcing. Please quantify your distinction in model performance.

Thank you for pointing this out, we will revise the text and quantify the performance of the models more accurately.

Line 329: "Despite cold conditions, an increase in snd still affects the snow insulation effect of LS3MIP CESM2." I don't understand the statement. Please reformulate.

What meant was that in the two categories (-25 to -15 and less than -25), the snow insulation effect shows an increasing trend with increasing snow depth at all depths. This phenomenon is not evident in Obs. Its been revised.

Line 336: I cannot follow this statement. Both in Wang et al 2016 and Burke et al 2020, previous versions of CLM5 (CLM45 stand alone in Wang et al, CLM4 in the CMIP5 analysis of CESM1 in the supplement of Burke et al) clearly outperform CLM5 with regard to the snow insulation curve. Please explain further what your statement is based on.

Burke et al. (2020) analyzed coupled models. And in Wang et al. (2016) the curve of category -5 °C to -15 °C is going up with a stronger trend than obs. Here we see a rather close distribution of snow insulation effect of CESM2-LS3MIP. We revised this paragraph by also discussing the results by Burke et al. (2020).

3.6 Impact of Land Model Features on Performance

Line 343: "show good performance in reproducing accurate snd" Actually, in Figure 2, the observed median value for snow depth is within the interquartile range of 1!! model in the LS3MIP forced simulations that supposedly do not suffer from biased precipitation. I would not call that good performance. Please add context.

It was wrongly calculated in Figure 4, we corrected the plot. While restructuring this subsection, we removed this paragraph.

Line 345: "Although IPSL-CM6A-LR employs a simpler spectral averaged albedo scheme than other land surface models, it does not have an observable impact on its tsl simulation." What data in your analysis is this statement based on?

In Figures 2, 3, and 4, the tsl of IPSL-CM is not worse than other models in terms of bias and spread. Only the bias in JJA is slightly higher, but it is still within 1 times the Obs IQR. However, we will phrase this claim more carefully.

Line 348: While vegetation is certainly important for accurately calculating albedo, in terms of the surface energy balance in general, the timing of snow cover is important. Please discuss the impact of a wrong timing of the onset of snow cover and melt.

We revised it. After consideration we are not discussing albedo in this section.

Line 349: "Considering snow conductivity, the Power Function could be why CNRM-CM6.1 and CNRM-ESM2.1 have a negative bias of larger than -6 °C in the SON (figure not shown)" In table2 , both the models with best snow insulation performance (the versions of JULES) and the model with the worst performance (Surfex) employ a power function, so this seems unlikely as the reason for the difference in performance. Please explain your conclusion in more detail.

In reviewing the references, we found conflicting information between Menard et al. (2021) and the primary references for JULES. Contrary to our initial understanding, both versions of JULES use a quadratic formulation of snow density to calculate thermal conductivity, as described in Wiltshire et al. (2020) and Calonne et al. (2011), rather than a power function. This correction requires a revision of our discussion, where we will re-evaluate the potential reasons for the observed biases in CNRM-CM6.1 and CNRM-ESM2.1.

Line 352: Especially in autumn, this could also be an effect of incorrect timing in snow. If snow cover is late in the models, the soil will release heat to the atmosphere for a prolonged time, which could also explain an underestimation of soil temperatures. Since you have not looked at the timing of snow cover, and snow rmse is large for all models in autumn at least in comparison to the stations considered in Figure 6, I think you need to extend your statement.

While splitting results and discussion, we removed this expression. Similar discussion about low snow depth is in section 4.6 but we are not talking about autumn snow as we could not identify the onset date of snow with monthly data.

Line 363: Since you cannot compare the performance of these models to versions that do not contain organic matter, I don't see how you can draw that conclusion. Please explain further.

We intended to refer to the "surface layer" rather than the "organic layer". In months

without snow cover, the surface insulation effects shown by the models are lower than the observed value, which is probably due to insufficient/missing representation of surface organic matter. Therefore, we made this assumption. But other factors like vegetation and model soil texture could also have impacts, so we will rethink and rephrase our explanation.

**4 Conclusions**

Please see my general comment on what the conclusion should contain. Specific comments below.

Line 373: "Except in summer months, inaccurate inter-annual variability in the simulation of soil temperature by CMIP6 models is mainly caused by deficiencies in the land surface models and less inherited from atmospheric components." What is the reasoning behind this conclusion?

As can be seen in Figure 5, the direction of model ensemble bias of CMIP6-tsl is commonly the same to LS3MIP-tsl but opposite to CMIP6-tas. And the ensemble standard deviation of LS3MIP-tsl is even larger than CMIP6-tas although LS3MIP is forced by same forcing, which indicates larger variability caused by the land surface model than the atmospheric model. We will add numbers here to support our conclusion.

Line 378: "The largest model biases of tas and tsl are witnessed under -5 °C." What does this refer to? Winter, summer, LS3MIP or CMIP6 models? And to what do the -5 °C refer? Climatological mean of winter temperature? MAGT? Please provide more context to explain the statement.

This conclusion is drawn from Figure 7/8/9. Regardless of season, when the temperature state itself is going under -5 °C, the standard deviation of Bias(model-Obs) grows. We will add more detailed explanation.

Line 379: "These indicate a weakness for models reproducing the tsl relationship with tas in freezing conditions" Which could point to deficiencies in soil moisture, which you have not discussed at all, even though it has a profound impact on latent heat during freeze and thaw. Please extend the discussion accodringly.

We will add in discussion section about the soil moisture.

Line 381: "Land models tend to simulate lower tsl when overlying snow exists." Do you mean lower than observed? Because as a general statement, that is wrong. Please explain more clearly.

Here the expression is not clear. We meant that the parametrization of snow insulation is insufficient for most models, so when there is snow, the model overestimate the energy loss from the land surface to the atmosphere. Causing the soil temperature to be lower than Obs under same tas/snd condition. We will rephrase.

Line 383: "Note that the scope of this study is limited to soil depths down to 0.2 m" You never state anywhere that all tsl metrics you show only refer to tsl at 20cm. Since the

RosHydroMet data provides temperatures at 20, 40, 80, 160 and 320 cm depth, I assumed all metrics referred to comparisons of all depths, and that only the snow insulation analysis is restricted to tsl in 20cm depth as proposed in Wang et al., 2016. This would have to to be clearly stated in the data description, but actually, I don't see any good reason for excluding the other depths from the general analysis, especially because you argue the relevance of the soil temperature analysis with the climate change impacts on permafrost, and 20cm depth is above the active layer thickness in large parts of the northern hemisphere permafrost area. Please extend the tsl analysis using all depths from the station data.

We will expand our scope (adding figures to appendix), and to make sure we have a clear statement. Still we will mostly focus on 20cm depth.

**For RC2:**

**Review of "Assessing Climate Modeling Uncertainties in the Siberian Frozen Soil Regions by Contrasting CMIP6 and LS3MIP"**

Firstly, I would like to commend the first reviewer for their thorough and detailed evaluation of this manuscript. The major strengths and particularly the weaknesses of the study have been clearly identified, leaving little room for additional comments. I fully agree with the assessment that this work would greatly benefit from major revisions to enhance its methodological rigour, clarity, and overall structure. Below, I provide a few additional comments that address secondary but potentially relevant concerns.

While this study provides valuable insights into the performance of CMIP6 and LS3MIP models in frozen soil regions, it would benefit from revisions to improve its methodological rigour, clarity, and overall structure. Removing redundant models, excluding ERA5-Land, providing a clearer discussion of results, and improving the presentation of figures would significantly enhance the manuscript's quality. I encourage the authors to consider these points in their revisions.

Thank you for your thoughtful review and constructive feedback. We appreciate your recognition of the insights provided by our study and the points raised for improvement. In response, we will carefully revise the manuscript to address the concerns listed below. Specifically, we will re-evaluate the inclusion of ERA5-Land, refine the discussion to ensure clearer communication of the results, and improve the presentation of the figures for better visual impact. We value your suggestions and will incorporate them to strengthen the overall quality of our work.

**1. Inclusion of Two Nearly Identical Models (CNRM-CM6-1 and CNRM-ESM2-1)**

One methodological issue that warrants attention is the inclusion of both CNRM-CM6-1 and CNRM-ESM2-1 in the analysis. These two models share an extremely similar structure and code base, making them effectively redundant. Their simultaneous inclusion biases the statistical assessment of model diversity and artificially reinforces certain trends. The authors should consider removing one of these models to ensure a more balanced and independent analysis.

We acknowledge the structural and codebase similarities between these models. However, we believe that analysis of both models also provides an opportunity to examine where their results diverge, which may provide additional insight into the impact of specific model components or configurations. Similarly, we note that the HadGEM and UKESM models are also closely related, with UKESM essentially being the ESM version of HadGEM. To address your concern, we will explicitly highlight the similarities between

these models and emphasize their differences in the relevant sections of the manuscript. In addition, we now adopt a weighting method proposed by Kuma et al. (2023). Consequently, Equation 3 in the manuscript is updated to read as follows:

$$EB_{i,s} = \sum_{m=1}^{M} \frac{1}{F\,N_m} \left( med_{m,i,s} - med_{o,i,s} \right)$$

where $m$ represents each climate model, $F=5$ is the total number of 'model families' and $N_m$ is the number of 'family members' within each family. In practice, in our case, this results in CESM2, IPSL-CM6A and MIROC6 being given weights of 0.2, while CNRM-CM, CNRM-ESM, HadGEM3 and UKESM are given weights of 0.1. This adjustment also affects Figures 5 and 9 in the manuscript.

**2. Questionable Use of ERA5-Land Data**

ERA5-Land is a reanalysis product, not an independent observational dataset. While reanalysis can sometimes provide useful large-scale validation, its role in this study appears unjustified. The manuscript already includes direct observations, which are far more suitable for model evaluation. Furthermore, comparing models against another model-based dataset (ERA5-Land) does not provide meaningful validation or evaluation. Removing ERA5-Land from the analysis would streamline the results and improve the manuscript's focus on actual observations.

We will no longer position ERA5-Land as a "benchmark" or reference dataset for model evaluation. Instead, ERA5-Land is presented as a widely used dataset that serves as a complementary basis for comparison. Since ERA5-Land was generated similarly to the LS3MIP simulations, a land-only model driven by reanalysis, ERA5-Land provides an opportunity to explore how reanalysis-based datasets perform alongside direct observational datasets. In addition, the ERA5-Land data have been regridded to match the resolution of the LS3MIP simulations (~0.9°), allowing us to assess the influence of resolution on the results.

**3. Overly Complex and Unreadable Figure 6**

Figure 6 is too dense and difficult to interpret, as it combines multiple variables (tas, tsl, pr, snd) in a single diagram. This makes it hard for the reader to extract meaningful insights. A better approach would be to separate this into multiple figures, each focusing on a single variable. For example, sub-figures or distinct panels could be used for each variable, with clear titles and well-defined legends. Additionally, clearer labelling and improved visual representation would greatly enhance readability.

A more effective approach would be perhaps to create a separate figure for each season, with four distinct panels for tas, tsl, pr, and snd. This structure would allow for a clear comparison of model performance across different seasons and variables, enhancing

readability and facilitating the identification of trends and anomalies. Using box plots or violin plots in each panel would effectively display the distribution of data, making the figures less cluttered and more insightful.

We will seperate the figure into two, one with tas and tsl, another with pr and snd (which will be put into appendix). However, we decided not to add more to the Taylor diagrams since they are still crowded.

**4. Lack of Discussion Section and Unstructured Conclusions**

As previously noted by the first reviewer, the manuscript lacks a dedicated discussion section, and its conclusions do not sufficiently synthesise the results in relation to the stated objectives. In particular, the authors should:

- Revisit the key research questions outlined in the introduction and explicitly address them in the conclusions.
- Provide a clear synthesis of the main findings, rather than scattering them throughout the results.
- Offer a more structured discussion, especially accounting for the following comments 5.

To address these issues, we will introduce a dedicated discussion section to thoroughly analyze the findings, situate them within the existing literature, and address their broader implications. In addition, we will revisit the key research questions outlined in the introduction and explicitly answer them in the conclusion section to ensure alignment with the study's objectives. The conclusions will also be restructured to clearly consolidate and synthesize the key findings and avoid the current scattered presentation.

**5. Lack of In-Depth Understanding of Model Processes and Literature Review**

One of the most concerning aspects of this manuscript is the apparent lack of in-depth understanding of the processes simulated by the six analysed models. Throughout the text, the authors make causal claims about model behaviour that are either too vague or lacking sufficient references, sometimes even incorrect, suggesting that they have not thoroughly studied the literature on these models. A deeper engagement with existing research would improve the accuracy of the study and prevent misleading conclusions.

Before attempting to diagnose model biases and uncertainties, the authors should conduct a more comprehensive literature review on each of the models they analyse. This would allow them to:

- Properly attribute biases to the correct physical processes,
- Avoid making incorrect causal inferences,
- Provide a more nuanced discussion of model differences.

A clear example of this issue is the discussion of CNRM-CM6-1 and CNRM-ESM2-1 in lines 350-353. The authors claim that the cold bias in these models is due to snow conductivity,

when in reality, it is mainly caused by the way these models simulate snow cover fraction as a function of vegetation (see section Snowpack Processes and Appendix B in Decharme et al. 2019). Unlike observations, which assume a fully snow-covered ground, these models allow for a snow-free fraction where soil is directly exposed to atmospheric forcing, leading to an artificially cold soil temperature. This is well-documented in the literature (e.g., Wang et al. 2016). For example, Decharme et al. (2019) states: "In addition, the specific snow fraction over tall vegetation is generally very low, annihilating the soil insulation effect of the snowpack." This explanation is completely absent from the manuscript, despite being a critical factor in the model's behaviour. Wang et al. (2016) also provide a robust and clear discussion of this problem in their "Model Processes" section.

In summary, the authors would benefit from reviewing Wang et al. (2016), which provides an excellent discussion of snow/temperature processes in models (see the "Model Processes" section, page 1733). Writing a discussion of equivalent quality about model processes is essential if this paper is ever to be accepted.

We recognize the need to strengthen our understanding and discussion of the processes simulated by the six models analyzed, and will address these concerns in our revision. We will conduct a more comprehensive literature review, including the work of Decharme et al. (2019) and Wang et al. (2016), as suggested. The discussion will be revised accordingly, and vague claims will be removed.

**For RC3:**

The manuscript addresses the persistent discrepancies in frozen soil simulations within climate models and their land components. By comparing historical runs from seven land-only models participating in the LS3MIP with their coupled counterparts in CMIP6, the study aims to disentangle the contributions of land surface parameterisations and atmospheric forcing to these discrepancies. Given the importance of accurate frozen soil simulations for climate projections and land-atmosphere interactions, this study aims to explore valuable insights into the strengths and limitations of current land surface models and their coupling with atmospheric components.

However, while the study's ambitions hold significant value for the scientific community, the execution falls short in several key areas. The manuscript lacks the clarity and rigor expected in an academic publication, with issues in writing style, structure, and methodological justification. Furthermore, some interpretations of model results are overly speculative, requiring a more cautious and evidence-based approach. Addressing these shortcomings through major revisions will be essential to ensure the study's findings are both robust and impactful.

We appreciate the reviewer's detailed feedback and constructive suggestions to improve the clarity and impact of the manuscript. We acknowledge the areas that require significant revision, particularly with regard to writing style and structure. We will carefully implement the recommended changes to ensure that the manuscript meets the standards of scientific publication.

**Major revisions**

- The writing style does not meet the standards of a scientific publication. Many sentences are overly long and vague, and overall consistency is lacking. Several sections—particularly parts of the introduction—are less rigorous and require significant clarification and development. To ensure the paper is accessible and acceptable to its readership, many statements need to be refined and expanded.

  Thank you for the feedback. We revised the manuscript to improve the writing style, making sure that the sentences are concise and clear and free of vagueness.

- The introduction requires a complete overhaul. I recommend restructuring it into distinct segments that provide:
- **Global/local context:** Outline the broader and specific contexts relevant to the paper.
- **Identification of issues:** Clearly identify and discuss the current challenges in frozen soil simulations.
- **Research question:** Present a specific, well-defined question that the study will address.
- **Study approach:** Detail how the paper intends to answer this question through its methods and analyses.

Thank you for your specific suggestions for restructuring the introduction. We rewrote the introduction part to make sure we consider the suggestions with better logic.

- The manuscript would benefit from a clear separation between the results and discussion sections. This division would enable readers to first digest the novel findings in the results section and then understand their interpretation and comparison with previous studies in the discussion. Currently, comparisons with existing literature are scant; the manuscript should incorporate and reference a broader range of studies relevant to the subject matter.

  We will provide separate results and discussion sections to ensure that readers can focus first on the results and then on their interpretation and contextualization. We will also place the results in the context of other relevant studies.

- Regarding the use of ERA5-Land data, the paper's central question is not directly related to the quality of this dataset. The statement "this proves that ERA5-Land can be a solid benchmark that supports observation as gridded data" is somewhat misleading. Providing "tas" values that closely match observations does not inherently qualify ERA5-Land as an appropriate benchmark for studies focusing on soil temperatures. Instead of justifying its use, the authors should expand the "Data and Methods" section and include a more thorough comparison with LS3MIP simulations to support their choice.

  We have revised our approach to the use of ERA5-Land data. We will no longer present ERA5-Land as a "benchmark" or reference dataset, but rather as a widely used dataset that provides an interesting basis for comparison. Similar to the LS3MIP simulations, ERA5-Land is a land-only model driven by reanalysis. In addition, the ERA5-Land data have been regridded to match the resolution of the LS3MIP simulations (~0.9°), allowing us to evaluate how the resolution affects the results. We will also expand the Data and Methods section accordingly.

- When results observations, such as biases or qualitative values (e.g., high, low, warm, cold) are mentioned, include specific numerical values. This practice will enhance clarity and allow for a more precise interpretation of the data. This

  We'll include specific numerical values in the results section to improve clarity and precision.

- Most importantly, the authors' attempt to link the biases observed in this manuscript to the physical processes represented or the parameterizations used in the models is largely misleading and lacks sufficient nuance. To make such claims, the authors could rely on sensitivity experiments, as they themselves acknowledge: "It is challenging to identify how model features influence the vertical energy transportation process without conducting sensitivity experiments." Without such analyses, the attribution of biases to specific model processes remains speculative. The manuscript would benefit from a more cautious interpretation of results, clearly distinguishing between observed discrepancies and their potential causes. Additionally, a more thorough review of existing literature on model parameterizations and process representations would provide a stronger

foundation for discussing the origins of biases. Instead of drawing causal conclusions prematurely, the authors should consider discussing alternative explanations, acknowledging uncertainties, and explicitly stating the limitations of their approach. For now, the results provided and their analysis are simply not sufficient to support the interpretations and conclusions.

In the revision we will take a more cautious approach, explicitly distinguishing observed biases from speculative causes. We will enhance the discussion by reviewing relevant literature on model parametrizations and process representations to provide better context. We will also clarify uncertainties and limitations in our methodology to ensure that interpretations remain balanced and well-founded.

**Figures and tables revisions**

- Verify and, if necessary, adjust the color palette to ensure it is color-blind friendly for all figures (I doubt figure 2/3 are for example).

  We tested the color blind readability of images on the inspection website suggested by The Cryosphere. The current palette is friendly to all color-weak users. However, since we used more than three models, we were unable to meet the reading requirements for red-green blindness.

Table 1:

- Specify which version of CLM5 is being used (e.g., CLM5.0).

  Yes, CLM5.0. Also adjusted everywhere else.

Table 2:

- Replace "snow conductivity" with "snow thermal conductivity." Note that all values are density-dependent except for MATSIRO. It would be more informative to share the default scheme used (e.g. Yen 1981)and reference the relevant publication rather than only providing the mathematical formulation.

  Initially we adopted the "wording" from Menard et al. (2021) and Wang et al. (2016), now we included the references of the used scheme as suggested.

Figure 1:

- Clearly indicate that "perma" refers to frozen soil at -20 cm, as opposed to permafrost at deeper layers elsewhere.

  We used a much strict way to select permafrost sites. As mentioned in the reply to RC1, the definition of permafrost in Lawrence and Slater, which requires at least two consecutive years of soil layers in a frozen state, poses two problems when implemented in this study. 1. Many sites lack deep soil layers below 0.8 m, and the soil above 0.8 m at these sites almost always thaws annually. 2. The study spans 30 years, during which some sites had frozen layers for two consecutive years, but the majority of the time was characterized by freeze-thaw cycles. We have therefore adopted a very strict permafrost screening scheme: a site is defined as permafrost only if the observation data at a depth of 3.2 meters (with sufficient observations)

remain completely frozen throughout the 30-year period. Of course, it is not rigorous to define only these sites as permafrost, so we will come back to the original definition when revising.

- Specify what is displayed by the color bar adjacent to the figure.

  We added info in the caption about the color bars.

- Consider adopting a visualization approach similar to that in Figure 5 for representing color classes—despite the current use of a linear color scale, note that employing too many classes (a maximum of 7×2 is recommended) can be problematic. If a diverging color scale is preferred, it should be centered around 0.

  We now calculate the difference from CMIP6 weighted multi-model ensemble to observation as the site color.

Figure 2:

- Ensure consistency across figures with respect to units, labels, and color schemes (e.g., using either "Celsius" or "°C" uniformly). The ERA5 Land color scheme should be consistent (currently it varies from gray to black), and model labels should be uniform. Choose ERA5 Land instead of ERA5_Land.

  We adjusted the figure according to the comment.

- The model labels and colors are not clearly aligned. It would be preferable for the legend to explicitly represent the color assignments and to differentiate offline land with a distinct style, as demonstrated in Figure 6.

  The labels are altered. But we don't consider now making offline boxes another style, it will add complexity to the readability.

- Clarify (in the figure caption and potentially also elsewhere in the text) that the site-average corresponds to the average of all selected station locations.

  Clarified in the caption, and information can be found in Data and Methods as well.

Figure 3:

- Ensure that the colors match exactly those used in Figure 2.

  Double checked, they are the same.

- The colors assigned to HadGEM and UKESM are too similar, making them difficult to distinguish.

  This is on purpose as they are using the land model from same family, just similar to CNRM models. We now use different markers for them, so they can be distinguishable.

Figure 4:

- There are no shaded areas.
- Include ERA5 Land in the legend.

  Fixed in caption and text (dashed lines instead of "shaded areas") and added ERA5

Land to the legend.

Figure 5:

- Align the orientation of both the figure and its caption.

  We use the template of EGU and assume that the Journal finalizes the layout.

- The current figure is challenging to interpret due to an overload of information. It would be beneficial to label each column or row directly within the figure, rather than forcing the reader to refer continuously to the caption.

  We improved the labels for each subplot, now it is straightforward for what is included.

- Include a legend that explains what the circle radii represent. Reducing the maximum radii is advisable to avoid overlapping circles, as both color and circle size are important for interpretation.
- Maintain a consistent cartographic projection for all maps; the projection used in Figure 5 currently differs from that in Figure 1.

  Valuable point, we will make them the same.

- Some data points are not legible. Although this may have been intended to portray small biases and standard deviations, it could be misinterpreted as insufficient data. Using an edge color (e.g., black) for the points may enhance clarity.

  The comments above are considered and fixed into Figure 5. We improved the labels and captions, so each map can be better understood what they are presenting. We also add the samples of sizes and edge colors to the circles.

Figure 6:

- The volume of information in this single figure could be overwhelming for readers. It is recommended to separate the diagrams by variable—focusing on the most relevant ones in the main text, while relegating the less critical diagrams to supplementary material.
- For enhanced clarity, consider dividing the legend into distinct sections: one indicating color (for different models), another for symbols (to distinguish between ESM and stand-alone LSM), and a third for numerical indicators (indicating variable significance).
- The "REF" is not adequately described; it should be clearly explained in the figure caption or within the main text.

  We will separate the four variables into two groups and present them across two distinct figures, while also modifying the legends for greater clarity.

  With regard to the term "REF", it is important to note that this refers to the point at which the 1 time normalized standard deviation, and the correlation coefficient to observation is 1. This description will be incorporated within the figure caption and discussed in the main text.

Figures 7 and 8:

- Although the manuscript states that "the model simulation outputs are binned at 2°C intervals," the x-axis labels and ticks reflect a 5°C interval.

  The data have been grouped into intervals of 2°C, as illustrated in the histogram pairs. However, the x-axis ticks have been set at 5°C intervals in order to accommodate the wide range of temperatures and to avoid the cluttering that would arise from using 2°C ticks.

Figure 9:

- The significance of the violin plots is not explained. The authors should clarify what the areas, dots, and black lines represent.

  We will enhance the figure by adding clear explanations of the violin plots, including what mentioned here.

- It remains unclear what the x-axis represents. Specify which "temperature" is being shown (e.g., air temperature or soil temperature, observations versus model results, and which dimensions are averaged).

  We will try to improve the style of the figure, and add more info via extra table or data in plot, to ensure that our statements can be easily understood. For your reference, the x-axis the categorized by observation temperatures (tas for the left subplot and tsl for the right subplot). We calculated 30-year mean at every month and station, so the data sample here has the size of 12month*236stations.

- The current style of the figure does not adequately support the subsequent discussion. For example, statements such as "Set Frozen and Set Warm tas data sample sizes are more than twice as large as in Set Intermediate" and "the difference in LS3MIP runs is negligible, suggesting that the climate model will likely have a cold and small deviation under these temperature states" are not easily decipherable. Focusing more on the standard deviation as a visual might improve interpretability.

  We added a table (Table 3) which supports the first statement and revised the text accordingly. The second statement was removed during revision.

Figure 10:

- It is surprising that this figure includes data from all seasons. Isn't it only winter data? If the intent is to follow the approach of Wang et al. (2016), the data should be restricted to winter conditions.

  Initially, the selection of DJF data was undertaken in accordance with the approach of Wang et al. (2016). However, due to an inadequate amount of samples for this period, the analysis was expanded to include data from other months as well. The present study focuses specifically on near-surface temperature data points below -5°C, utilizing monthly average values. Consequently, it is hypothesized that the prevailing selection criteria can provide a satisfactory and reliable sample size for analysis.

- Position the model names at the top of each panel to enhance readability.

  Model names have been added to the top of each sub-figure.

**Minor revisions**

Thank you for your valuable and specific comments, which we greatly appreciate. We will take them into account in our revisions to improve the manuscript. In particular, the introduction will be rewritten and the results and discussion sections will be separated. Below, we address some of your comments individually where further clarification or alternative perspectives are needed; for other comments, we agree with your suggestions and will revise accordingly.

General

- Some terms need to be more explicitly defined:
- **Models:** Clearly define which type of models is under discussion (e.g., Earth System Models, land surface models, etc.) in the introduction.

  We will make brief introduction of Climate Model, Earth System Model and Land Surface Models in the introduction.

- **Model Ensembles:** When introducing model ensembles, particularly in reference to CMIP6 and LS3MIP, specify this term explicitly.

  We will revise the text to explicitly define and specify the term "model ensembles" in the context of CMIP6 and LS3MIP for clarity and precision.

- **Land-Only vs. Offline land surface models:** Adopt and consistently use one term throughout the manuscript.

  We use term "land-only".

- **Permafrost:** At line 152, the manuscript classifies certain locations as permafrost, which may be misleading because it may imply that permafrost is present exclusively at these sites. The explanation provided in lines 223–227, clarifying that these locations exhibit permafrost soils at a depth of –20 cm, should be introduced earlier and applied consistently to avoid ambiguity

  The classification problem is answered above in Figure 1 comment. Here, this was an error in expression. We will clarify a solution and maintain consistency throughout the text.

- Use negative numbers consistently when referring to depth and cold bias values. For example, if cold biases are sometimes shown as negative values, ensure that all such instances follow that convention.

  We corrected phrases with misleading signs regarding cold biases.

- The authors are encouraged to provide the code used to produce the figures in addition to the underlying data. This will improve transparency, reproducibility, and the ability for readers to further explore and validate the results.

In the revised manuscript, we will provide a link to Zenodo where we will upload the relevant scripts.

Introduction

- **Lines 20, 44, 46, 78, 94–97:** The authors should include additional references at these lines.

  We have rewrote the introduction and added new references.

- **Lines 21, 171:** Update the references with more recent publications to reflect the current state of research.

  Line 21 updated, see first paragraph of Introduction. We rewrote the results and discussions, and removed line 171.

- **Line 24:** The sentence is overly long and should be rewritten into two or more concise sentences to improve clarity and readability.

  Resolved during rewrite of the introduction.

- **Line 28:** If abrupt thaw is mentioned, a clarification is needed to explain its relevance to this study. The authors should clearly delineate the connection between abrupt thaw processes and the objectives of the manuscript.

  Our research does not specifically address abrupt thaw. It is mentioned here to illustrate the importance of permafrost research in the context of climate change. We will consider reducing this content or expressing it more logically and coherently.

- **Line 29:** Clarify the phrase "such carbon emissions" by specifying what these emissions refer to, ensuring that the meaning is unambiguous.

  Removed this phrase during rewrite of the introduction.

- **Lines 30–32:** The implications of "changes in surface vegetation types" should be further developed. While the focus on permafrost thaw is noted, the authors need to expand the discussion to include other consequences of permafrost thaw on Earth's ecosystems, not just those related to vegetation.

  We modified this into: "*Permafrost thaw alters hydrothermal conditions, which can alter surface vegetation depending on soil moisture. In lowland regions with ice-rich permafrost, abrupt thawing is often followed by vegetation recovery. Under stronger or prolonged changes, the system may reach a tipping point, beyond which widespread ecosystem disruption can occur (Heijmans et al., 2022)*"

- **Lines 34–35:** Revise the sentence for clarity. The term "frequently varying" is ambiguous, and the concept of "thermal offset" should be explicitly defined and contextualized.

  Revised as: "*Model parameters for hydrothermal transport are governed by soil texture, surface organic matter, moisture dynamics, and freeze–thaw conditions (Woo, 2012; Andresen et al., 2020; Yang et al., 2022). In permafrost regions, these factors determine the thermal offset (Kudryavtsev, V.A., 1977)—the temperature*

*difference between the ground surface and the top of the permafrost—by altering soil hydrothermal properties.*"

- **Lines 35–37:** The statement regarding differences in time scales between soil and atmospheric processes and the role of the soil surface as the interaction window needs further refinement. A clearer explanation of these dynamics, with concrete examples if possible, will help strengthen the argument.

  We extended on that in the 4th paragraph in the introduction.

- **Lines 36–38:** Should examples be mentioned here, the authors are advised to include at least a couple of specific cases. For instance, emphasizing "excess ice" conditions (as noted by Burke et al. (2020) and other studies) would help illustrate the point effectively.

  Revised as: "*Incorporating soil ice and water dynamics into land surface models improves simulations of active layer hydrothermal conditions by capturing seasonal freeze–thaw processes and moisture effects, such as summer cooling and winter warming of permafrost due to increased soil moisture (Swenson et al., 2012; Li et al., 2021; Du et al., 2023). Including soil ice dynamics in models allows for the simulation of ice-wedge degradation and associated ground subsidence, capturing rapid landscape changes such as thermokarst under strong warming scenarios(Liljedahl et al., 2016; Nitzbon et al., 2020; Cai et al., 2020).*"

- **Line 41:** The statement is currently too vague. The authors should detail what each mentioned characteristic does and how it impacts the study.

  The following statement is added: "*A thick snowpack provides stronger thermal insulation, which limits soil heat loss in winter and delays thawing in spring. Lower-density snow insulates more effectively due to its reduced thermal conductivity. The duration of snow cover determines the length of the insulated period, which affects the timing and amplitude of seasonal soil temperature changes.*"

- **Line 42:** Specify which conditions are being referred to (including the time-scale, any specific event, and the relevant soil depth), providing the reader with necessary context.

  Revised as: "*Research has shown that changes in snow conditions (snow depth, density, and duration) account for over 50% of variations in soil temperatures observed in northeastern Siberia  (Park et al., 2014, 2015).*"

- **Line 44:** The phrase "the most suitable" is subjective and should be replaced with more objective language.

  Rephrased and reduced wordiness: "*It allows evaluation of the ability of the latest generation of climate models to simulate frozen soil ...*"

- **Line 45:** The text mentions horizontal resolutions; it should be clarified why high resolution is necessary to distinguish between different frozen soil regions. A brief explanation or supporting evidence is recommended.

  Explanation added as: "*The extent and characteristics of frozen soil can vary abruptly over short distances, especially in complex terrain or transition zones*

*between different types of permafrost."*

- **Line 47:** Provide further justification for the necessity of high-resolution data at this point in the paper.

  We added explanation above. While we acknowledge that spatial resolution plays a role in representing permafrost heterogeneity, the range of horizontal resolutions used in CMIP6 models reflects a balance between realism and computational feasibility. Within this resolution range, previous studies have shown that further refinement yields limited improvements in large-scale soil thermal dynamics relative to the substantial increase in computational cost. A discussion of resolution impacts regarding ERA5-Land is provided in the Discussion section.

- **Line 56:** Describe the potential consequences (or cite relevant studies) that are being discussed. It is advisable to move this sentence to an earlier position in the introduction, before the datasets are introduced, to frame the context properly.

  We cited two relevant studies (Schuur et al., 2015; Streletskiy et al., 2025) to support this statement.

- **Line 59:** This sentence is overly long.

  We split the sentence as: "*Within this region, the observational dataset provided by the All-Russian Scientific Research Institute of Hydrometeorological Information-World Data Center (RIHMI-WDC) can be used. This dataset provides consistent soil temperature measurements at standardized depths and can thus be used as a reference in climate model evaluation.*".

- **Lines 61–62:** The delineation of which "characteristics" are being assessed is too vague. In addition, the concept of a "benchmark" is not developed. The authors should specify which benchmark is being applied.

  It is further described in the following sentences, which we also refined. We replaced "benchmark" with "references" at this point.

- **Lines 62–64:** Although the general concept is sound, this section should be expanded. The authors need to elaborate on the differences between CMIP6 and LS3MIP that could lead to the observed biases and uncertainties in frozen soil regions. In particular, clarify how these differences might be attributed to discrepancies arising from the land surface models versus those caused by atmospheric forcings.

  We expanded in the following sentences: *"Under identical, observation-based atmospheric conditions, the LS3MIP models are expected to simulate soil temperature more accurately than their CMIP6 counterparts. If not, discrepancies may indicate limitations within the land surface models themselves, such as deficiencies in parameterization or missing processes that impair their ability to respond appropriately to atmospheric forcing. Conversely, errors found in coupled CMIP6 simulations may result from biases in atmospheric forcing, such as misrepresentation of near-surface air temperature, precipitation, or surface radiation. This experimental design allows us to distinguish the sources of uncertainty between land-only and coupled simulations. Additionally, we will*

*explore inter-model variations within LS3MIP and assess how specific structural features, such as bottom boundary conditions and snow thermal conductivity parameterizations, relate to model performance in frozen soil regions."*

- **Line 64:** The phrase "with identical and more realistic atmospheric conditions" is ambiguous. Clarification is needed regarding what is meant by this and why, under such conditions, LS3MIP models are anticipated to simulate soil conditions more accurately.

  Changed "more realistic" to "observation-based". And "soil conditions" to "soil temperature"

- **Lines 65–66:** This sentence, which is key to the study, requires further development. The rationale behind regarding certain discrepancies as "errors in the land surface models" needs to be clearly explained, with supporting arguments that make the rationale accessible to all readers.

  We revised as written above.

- **Line 67:** Clearly specify which features are being referred to. Providing examples where applicable will help avoid ambiguity.

  We specifically added the discussed features here, as written above.

Data and methods

- **Line 76:** Consider introducing the concept of climate "feedback" in the introduction, as it is a key component of this study. This will help set the stage for its later use in the analysis.

  We mentioned in third paragraph of introduction, about climate feedback. However, it is not further discussed in our study.

- **Lines 78–80:** The sentence in these lines is unclear. A revision is needed to improve clarity and ensure that the intended meaning is conveyed unambiguously.

  Revised as "*This setup allowed us to directly compare CMIP6 and LS3MIP results and disentangle the relative contributions of coupling-related errors and land model deficiencies to biases in frozen soil regions.*"

- **Line 81:** Replace "climate models/earth system models" with "Earth System Models (ESMs)" to maintain consistency and accuracy in terminology.

  We removed the term "earth system models" and kept the more general term "climate models". ESM would not apply to CNRM-CM6.1, MIROC6 , and HadGEM3 (e.g. see definition by Eyring et al., 2016 or the nice overview graphic from Kuma et al., 2023). So we keep the more general term "climate models" throughout the manuscript.

- **Lines 82–83:** Rephrase "cannot be considered" to clarify whether the models lack a freeze option when turned off or if they do not adequately represent frozen soil processes.

  We revise the sentence as *"Other climate models, which also participated in both*

*projects, are not included in this study as they either turned off the freeze option in frozen soil in the CMIP6 version or did not provide data for all our target variables."*

- **Line 84:** Remove any repetitive wording to improve the flow of the section.

  Revised as *"Hereafter, we refer to the CMIP6 as Group C and the LS3MIP as Group L in plots and analysis.".*

- **Line 91:** Use the term "snow thermal conductivity".

  Revised.

- **Lines 91–92:** The current description is somewhat misleading. It should be clarified that all formulations are either (1) empirically derived and density-dependent or (2) assigned fixed values. This nuance is important for understanding the parameterizations.

  We removed albedo and rewrote this part to *"The thermal conductivity of snow is modeled using density-dependent formulations as a power function (Yen, 1981) or a quadratic function (Jordan, 1991; Calonne et al., 2011; Wang et al., 2013), or using fixed values."*

- **Line 100:** Replace the vague phrase "assist our assessment" with a more precise description of how the method contributes to the analysis. Additionally, the term "numerical" is too vague; the authors need to clarify and explain the differences between ERA5-Land and other land surface models, including a brief definition of what reanalyses entail.

  We removed this sentence as we do not use ERA5-Land as benchmark.

- **Line 103:** Provide details on how the available depth data are interpolated to the target depth of the study. This clarification is necessary for understanding the data processing methodology.

  We added the info that we interpolated linearly between depths.

- **Line 107:** Specify which quality flag is being referenced and explain what it represents regarding data quality or processing.

  Info added.

- **Line 108:** Reconsider the rationale for using user-defined values of longitude and latitude to determine warmer climates. Note that areas east of 120°E do not correspond to Siberia. It may be preferable to use average air surface temperature measurements to define these regions.

  We revised the expression to *"Stations west of 60°E, east of 120°E, and south of 45°N are excluded to eliminate most stations with warmer climates. Moreover, sites are classified based on their winter 2 m air temperature."*

- **Line 128:** The phrase "in the central tendency of the data" is ambiguous.

  Revised as *"…addresses systematic deviations using the median data values."*

- **Line 139:** Clearly define the seasons used in the analysis. For example, if DJF is employed, specify whether it covers December 1st to February 28/29th or follows another seasonal definition.

We added a clarification how we define the seasons.

Results

- **Line 142:** Instead of beginning every sentence with "Fig. x…", the authors should directly present the scientific point. This repetition occurs multiple times and could be streamlined to improve readability.

  We rephrased most of the sentences starting with "Fig…".

- **Line 143:** The term "matching" is superfluous and should be removed.

  Removed.

- **Lines 147–148:** Clarify the rationale behind the observations made here. Consider moving this explanation to an earlier portion of the section so that the context is established before the results are discussed.

  Now we put this into the discussion section.

- **Line 150:** The term "outcomes" is vague.

  Changed to "results".

- **Line 162:** The term "interpolated" is ambiguous. The authors should specify the interpolation method used and indicate how this process might affect the results.

  Replaced with "derived". We stay with the more general wording here and do not further specify the interpolation methods. Our goal is not to discuss the differences of *tas* and *pr* between the LS3MIP models in detail, but rather compare with thier corresponding CMIP6 model.

- **Lines 162–163:** The sentence stating, "Fig. 2 shows slight differences between different land models because of interpolation uncertainties using different model grids with different setups," uses the terms "differences" and "different" in a repetitive way. The authors should (a) avoid rushing to conclusions by providing supporting quantitative evidence (e.g., specific values or statistical measures), and (b) rephrase the sentence to clearly explain the potential impact of grid differences and interpolation uncertainties.

  We split Results and Discussion, added numerical values, and rephrased to "However, Fig.2 shows slight discrepancies of tas (less than 1 °C) and pr (less than 3 mm) among land-only models." and "Even though the LS3MIP models were forced with identical reanalysis data, their tas and pr medians differed slightly, as they have different grid resolutions and methods for calculating and regridding tas and pr."

- **Lines 163–164:** The statement, "This illustrates how carefully a comparison of coarse-grid model output against point-like station data has to be interpreted," requires further explanation. The authors should provide concrete evidence or reasoning to demonstrate how this conclusion was reached.

  Revised as: "*This emphasizes the importance of recognizing that gridded model outputs are spatial averages constrained by grid cell resolutions. Station*

*observations, on the other hand, may not represent the broader grid-scale climate. Therefore, higher consistency between the two does not necessarily imply better model performance. Furthermore, discrepancies in temporal variability between modeled and observed climate can complicate interpretation when comparing with CMIP6 outputs. In this study, we focused on evaluating the models' ability to simulate snd and tsl with similar atmospheric forcing (tas and pr).*"

- **Lines 170–171:** Rephrase and clarify the material within the brackets to ensure that it is concise and informative.

  We put the content in the brackets into a separate sentence: "The CNRM simulations exhibited the lowest soil temperatures overall (Fig.2) with the climatologies of being more than 8C lower in both CMIP6 and LS3MIP."

- **Line 171:** The phrase "same family" should be explicitly defined. The authors need to clarify what criteria determine the grouping of models into the "same family."

  We added the definition of "model family" in Methods, following the definition by Kuma et al. (2023).

- **Line 172:** The phrase "their ability to simulate tsl" is too vague. The authors should detail why a particular performance in simulating soil temperature (tsl) is expected and what underlying processes or parameterizations support this expectation.

  We removed this phrase. In the new discussion section we discuss the results regarding some model features.

- **Line 174:** I am not a statistician, but the term "diversity" is not adequate in this context and in the rest of the manuscript for me. Could it be replaced with a more specific term such as "variability" or "spread" in model performance to better describe the differences observed?

  Yes, using "diversity" here is inappropriate. We will change it to "spread."

- **Line 175:** Clarify what is meant by "with most sites lacking insulating snow." The authors should specify the criteria or observations underpinning this statement and discuss how this influences the results.

  Revised as: "...*particularly as snow was absent at most sites during summer (median value less than 0.3 cm)*"

- **Line 177:** Specify which models are being referred to at this point to avoid ambiguity.

  This content will be discussed in the next section so we removed the sentence from here.

- **Lines 207–208:** The sentence is redundant or not essential to the discussion.

  We removed this sentence.

- **Line 211:** The assertion that "Differences in grid cell scale among models can lead to biases in the tas state over the grid" is unclear. The authors should expand on this point—explaining how grid cell scale differences can affect biases in near-surface air temperature (tas)—and support the statement with references to the

literature or numerical values.

This is actually a minor effect, as the resolution will impact the grid included for site interpolation, and grid area differences are the source of biases in near-surface air temperature. We removed this assertion.

- **Lines 218–220:** While the discussion of differences in tas EB and tsl EB between LS3MIP and CMIP6 simulations offers an interesting perspective regarding the compensation between land surface and atmospheric processes, this section needs further clarification. The authors should:
- Provide explicit numerical values or clear graphical support for the claim.

We provided more data (numbers) to support our analyses.

- Reconcile this discussion with other observations in the manuscript (e.g., the statement in line 234 regarding Group L versus Group C performance).
- Elaborate on the physical reasoning behind the offsetting errors observed in the CMIP6 ensemble.

The compensation phenomenon are further discussed in Discussion section Climate Dependency of Modeled Temperatures.

- **Line 223:** The focus on the "shallow soil response" should be clarified and introduced earlier in the section or even in the introduction to offer better context to the reader.

Moved to Introduction.

- **Line 230:** The use of the term "better" is vague. A more precise descriptor or quantitative measure of comparative performance should replace it.

We rewrote this section and avoided using "better" in this case.

- **Line 234:** It is unclear whether the reported tsl values pertain to all seasons. The authors should clearly state which seasons are included and, if possible, quantify the differences observed.

We rewrote this section as the classic definition of Permafrost was applied, and added numerical values to better illustrate the figure.

- **"Climate Dependency of Modeled Temperatures" Section:** This entire section could be confusing as it references two kinds of "temperature values":
1. Cold/warm biases (often without specific numerical values)
2. Temperature values from observations or model outputs (without clear designation) It is recommended that the authors adopt a consistent approach similar to that used by Wang et al. (2016) and further in the manuscript, where temperature regimes are clearly defined and specific numerical ranges are provided for each regime.

We revised this section to distinguish two temperature values.

- **Line 243:** The term "state" needs to be clearly defined. Furthermore, introductory words such as "So," (and "And" further in manscrupit) should be removed in favor of a more formal tone. This writing style is very very preoccupying.

We will try to remove/revise such expressions to make the text more formal.

- **Line 247:** The phrase "simulate similar histograms" is misleading because histograms represent sample counts.

  Revised as: "*...reproduced the observed sample distribution...*"

- **Line 248:** The statement "However, a slight cold bias below -30 °C exists in Group L" requires clarification. If such a bias is observed, the authors should provide the exact numerical value, discuss its significance, and ensure the figure clearly demonstrates this bias.

  This was misleading, we indicates there was a cold bias when temperature was below  -30 °C, but not bias is  -30 °C. We refined such expressions.

- **Line 261:** The phrase "more likely distributed" is unclear. The authors should:
- Specify how the distribution of tsl was characterized.
- Indicate the specific temperature range of the cold bias.

  We use now "sample distribution" to make it clear that distribution is defined by the samples counts in different intervals. And we revised corresponding texts to make the temperature ranges clear.

- **Lines 266–268:** This passage needs to be rewritten for clarity. For instance:
- Clearly specify that the values refer to the minimum tas (if that is the case).
- Rewrite "than that of the land-only run."
- Explicitly describe how differences in the lower extremes of tas translate into corresponding gaps in tsl values, supporting this statement with numerical evidence.

  We rewrote the paragraph to address the comments. See Lines 323-326.

- **Line 269:** There is a typographical error: "underestimate" should be corrected to "underestimating."

  We rewrote as in the first paragraph of Sect.4.4.

- **Lines 269–270:** The sentence is overly long and ambiguous. It should be restructured to clarify the differences between the "snow insulation effect" and the "surface insulation effect". Also, note that Dutch et al. (2022) discuss only CLM5.0. The authors should clearly explain whether the two effects are distinct or closely linked, and provide additional literature to support any claims regarding deficiencies in the modeled snow insulation effect.

  We rewrote this part as mentioned above.

- **Line 272:** Missing the term "flux".

  Added.

- **Lines 273–274:** The claim—that a decrease in tas has a limited influence on tsl due to high snd, implying that the primary source of error stems from thermal conditions at the bottom of the soil column—requires stronger substantiation. More data, evidence, or references should be provided to support the assertion regarding the role of geothermal flux.

*This claim was not precise, we explained as: "This phenomenon exhibited strong model dependence, suggesting that the cause was not only related to insufficient representation of surface and snow insulation but may also stem from factors such as overly strong thermal resistance in the land-only model parameterizations or low soil moisture content."*

- **Lines 277–280:** The categorization of states (i.e., the thawed state, the freeze–thaw transition state, and the frozen state) needs further clarification. It is recommended to (a) reference literature that has adopted a similar categorization technique and (b) ensure consistency in nomenclature between the text and figures—for example, aligning terms like "Set Frozen, Set Intermediate, Set Warm" with the descriptive categories.

  We want to make sure that in set frozen and set warm the soil in observation is completely frozen or thawed. That's why we choose -5 °C and 5 °C as the boundary.

- **Line 282:** Remove "Results are shown in Fig. 9".

  Removed.

- **Line 287:** Clarify whether the observation that "the tsl samples are mainly concentrated in Set Intermediate and Set Warm" is directly evident from Fig. 9. If not, provide either numerical summaries or additional explanation within the text or figure caption.

  We added Table 3 to add the necessary numbers.

- **Lines 290–292:** The claim that "the mean and minimum value of tsl bias is much lower than that of tas bias" and that this negative bias appears across all sets requires further explanation. The authors should provide precise numerical values and discuss how these figures support the conclusion that the land models simulate tsl as being too cold relative to expectations.

  The numbers are now be listed in Table 3. We removed 'the land models simulate tsl as being too cold relative to expectations' as this was not statistically right after family weighting. Whats true is that tsl simulation are highly diversed in Set Frozen.

- **Lines 292–293:** The key point that improved tas accuracy in Group L models (in Set Frozen and Set Intermediate) does not necessarily yield better tsl simulations, and that tsl variability below –5 °C is higher in Group L than in Group C, needs to be expanded.

  We discussed about it in Lines 453-455

- **Line 303:** The statement that "the tsl gradually convergences near 0 °C" is not clearly supported by the observation figure. In addition, the phrase "and is primarily impacted by tas in a limited manner" appears contradictory. The authors should re-examine the data, reconcile these inconsistencies (especially in light of their earlier remark in line 273), and clearly articulate the influence of tas on tsl with supporting evidence.

  We corrected and rephrased this part.

- **Lines 305–307:** The discussion of snow shielding effects—claiming that thicker snow (snd > 0.3 m) strongly relates ΔT and tas—needs to be clarified. The authors should elaborate on how thicker snow modifies the impact of overlying tas and provide additional data or references to substantiate this relationship.

  We have clarified by elaborating on the insulating effect of snow due to its low thermal conductivity.

- **Lines 316–317:** The claim that "all other models fail to reproduce the observation-like curve, underestimating the snow insulation effect under most conditions" seems overly generalized. For instance, CESM2 appears to capture the curve reasonably well. The discussion should differentiate among models and provide detailed evidence to support such claims.

  We described it a bit more fine-grained now. However, more discussion on CESM, CNRM, and MIROC was provided two paragraphs below.

- **Lines 319–321:** The explanation that low-resolution land surface models hinder accurate determination of surface organic matter distribution—thereby leading to errors in calculating the surface insulation effect—requires further detail. The authors should clarify the underlying processes, reference additional studies (e.g. 10.1175/JCLI-D-24-0267.1), and distinguish this issue from concerns related to snow insulation.

  We will rewrite this part, focus mainly on snow insulation. The function of soil organic matter will be briefly discussed in last section of discussion.

- **Lines 319–324:** This section appears to focus on surface insulation rather than directly addressing snow insulation. Given that other studies (e.g., 10.1038/s41467-019-11103-1) have highlighted that the warming effect of soil organic matter is less significant in winter because of insulating snow cover, it might be advisable to revise or remove this passage.

  The same as above.

- **Line 325:** The statement "Similar conclusions can be made to HadGEM3" should be clarified by explicitly detailing which aspects of the analysis are similar and providing supporting evidence for this comparison.
- We assume you mean UKESM in this line. We extended this sentence and clarified what me mean by both ensembles (CMIP6 vs. LS3MIP).
- **Lines 334–340:** More numerical evidence is needed to support the claims made in this portion of the discussion. The authors should include specific numbers, statistical measures, or comparisons to strengthen their argument.

  We revised this paragraph by also discussing the results by Burke et al. (2020) and added numerical values in a Table and in the text for more direct comparison with Wang et al. (2016).

- **Line 340:** The phrasing "where a substantial reduction in the lack of snow insulation is seen" contains a confusing double negative.

  Simplified the two sentences into "Compared to its previous version, ORCHIDEE (IPSL-CM6A-LR) exhibits a larger snow insulation effect that is closer to observed

values."

- **Lines 341–368:** There is a lack of quantitative support throughout this portion. It is recommended to supplement the discussion with numerical values that back up the claims. In particular, when addressing snow thermal conductivity, compare the performance of different parameterization schemes (rather than solely presenting their mathematical formulations) and clarify the impact of low snow depth on thermal behavior.

  We revised this section by comparing and discussing the different parametrizations for snow thermal conductivity more in detail.

Conclusions

- **Lines 373–382:** The authors need to substantially develop and clarify nearly every sentence in this section. Several statements are not consistently supported by the manuscript. For example:
- The last figure concerning CESM shows results that contradict the claim that "inaccurate inter-annual variability in the simulation of soil temperature by CMIP6 models is mainly caused by deficiencies in the land surface models and less inherited from atmospheric components." This discrepancy is not adequately addressed in the discussion.

  We revised the Conclusion section for better logic. And a separate Discussion section is written to clarify the conclusions we draw.

- The statement that "biases in the land surface model even partially compensate for the influence of air temperature biases" lacks sufficient evidence or numerical backing.

  We added numerical values in results and discussions to support this conclusion.

- The claim that "better precipitation simulation does not ensure snow depth results improve, especially in winter and spring" is not clearly linked to the rest of the manuscript.

  We added it in the last paragraph of Sect. 3.3

- Terms such as "weakness," "near-surface energy transport process," and "snow amount" are imprecise.

  Changed to "limitation" and rephrased the other sentence.

- The recommendation for "further improvement of parameterization" is vague. The authors should identify which specific parameterizations (e.g., those related to snow dynamics, soil thermal properties, or energy exchange processes) require refinement.

  We claimed this recommendation in 3 paragraphs (Lines 532 to 555) in Conclusion and listed them in detail.

**References**

Cai, Z., You, Q., Wu, F., Chen, H. W., Chen, D., & Cohen, J. (2021). Arctic warming revealed by multiple CMIP6 models: Evaluation of historical simulations and quantification of future projection uncertainties. Journal of Climate, 34(12), 4871-4892.

Calonne, N., Flin, F., Morin, S., Lesaffre, B., du Roscoat, S. R., & Geindreau, C. (2011). Numerical and experimental investigations of the effective thermal conductivity of snow. Geophysical research letters, 38(23). https://doi.org/10.1029/2011GL049234

Eyring, V., Bony, S., Meehl, G. A., Senior, C. A., Stevens, B., Stouffer, R. J., & Taylor, K. E. (2016). Overview of the Coupled Model Intercomparison Project Phase 6 (CMIP6) experimental design and organization. *Geoscientific Model Development*, 9(5), 1937-1958. https://doi.org/10.5194/gmd-9-1937-2016

Menard, C. B., Essery, R., Krinner, G., Arduini, G., Bartlett, P., Boone, A., Brutel-Vuilmet, C., Burke, E., Cuntz, M., Dai, Y., Decharme, B., Dutra, E., Fang, X., Fierz, C., Gusev, Y., Hagemann, S., Haverd, V., Kim, H., Lafaysse, M., Marke, T., Nasonova, O., Nitta, T., Niwano, M., Pomeroy, J., Schädler, G., Semenov, V. A., Smirnova, T., Strasser, U., Swenson, S., Turkov, D., Wever, N., and Yuan, H.: Scientific and Human Errors in a Snow Model Intercomparison, *Bulletin of the American Meteorological Society*, 102, E61–E79, https://doi.org/10.1175/BAMS-D-19-0329.1, 2021.

Koven, C. D., Riley, W. J., & Stern, A. (2013). Analysis of permafrost thermal dynamics and response to climate change in the CMIP5 Earth System Models. Journal of climate, 26(6), 1877-1900.

Kuma, P., Bender, F. A. M., & Jönsson, A. R. (2023). Climate model code genealogy and its relation to climate feedbacks and sensitivity. Journal of Advances in Modeling Earth Systems, 15(7), e2022MS003588.

Wang, W., Rinke, A., Moore, J. C., Ji, D., Cui, X., Peng, S., Lawrence, D. M., McGuire, A. D., Burke, E. J., Chen, X., Decharme, B., Koven, C., MacDougall, A., Saito, K., Zhang, W., Alkama, R., Bohn, T. J., Ciais, P., Delire, C., Gouttevin, I., Hajima, T., Krinner, G., Lettenmaier, D. P., Miller, P. A., Smith, B., Sueyoshi, T., and Sherstiukov, A. B.: Evaluation of air–soil temperature relationships simulated by land surface models during winter across the permafrost region, The Cryosphere, 10, 1721–1737, https://doi.org/10.5194/tc-10-1721-2016, 2016.

Wiltshire, A. J., Duran Rojas, M. C., Edwards, J. M., Gedney, N., Harper, A. B., Hartley, A. J., Hendry, M. A., Robertson, E., and Smout-Day, K.: JULES-GL7: the Global Land configuration of the Joint UK Land Environment Simulator version 7.0 and 7.2, Geosci. Model Dev., 13, 483–505, https://doi.org/10.5194/gmd-13-483-2020, 2020.

---

## Author Response (AR2)

We would like to thank the reviewers for their detailed and comprehensive comments and suggestions. We greatly appreciate the time and effort you have devoted to reviewing our manuscript. Below we respond to each comment point by point. The reviewers' original comments are in **black**, and our responses are in **blue**.

**For RC2:**

General comment

The revised manuscript shows clear improvements in structure and clarity, and ERA5-Land is now appropriately positioned as a complementary dataset. The introduction of model-family weighting also helps to reduce the over-representation of certain models. However, several figures (e.g., Figs. 5 and 6) remain visually dense, and the discussion retains a somewhat descriptive tone. Most importantly, the attribution of biases to specific physical processes in the models remains incomplete, with important explanations from the literature still missing. I therefore consider that the paper requires minor revisions before publication.

1. Inclusion of CNRM-CM6.1 and CNRM-ESM2.1

The authors have introduced model-family weighting to reduce the over-representation of the two CNRM versions. This change is a step in the right direction. Thank you.

2. ERA5-Land

ERA5-Land is no longer presented as a primary evaluation benchmark but as a complementary dataset, with a clear explanation of the regridding and its limitations. This fully addresses the initial concern, and no further adjustment is needed on this point.

3. Figures 5 and 6 too dense

Figure 6 has been partially simplified: tas/tsl variables are kept in the main figure, while pr/snd have been moved to the appendix, and graphic elements have been enlarged for better readability. However, the main figure still appears dense, with many overlapping symbols and layers of information, which limits clarity. Further simplification could improve understanding. The same applies to the new Figure 5.

For Figure 5 we now only show DJF results in the results section, which allows us to enlarge the figure, and put the original figure to the appendix, which includes all seasons.
For Figure 6, we restructured the labels in the bottom and enlarged the main figure. The label used for CMIP6 models are altered now to open circles instead of bullets,

thus the overlapping are reduced.

4. Discussion and conclusions

The revised manuscript now includes a dedicated Discussion section, organised into subsections that address the objectives set out in the introduction, as well as a more structured conclusion. This improves the link between objectives, results, and interpretations. However, it would be beneficial to include more perspectives on the implications for permafrost or climate modelling and to reduce the repetition of results.

We rewrote most of the Discussion section to reduce repetitions and added more supporting points.

5. Understanding of processes and literature review

A factual error regarding the formulation of snow thermal conductivity in JULES has been corrected, which is a positive step. Nevertheless, the main explanation for the cold bias in the CNRM models—linked to the representation of the snow-free fraction beneath tall vegetation—remains absent or superficially addressed, despite references available in the literature (Decharme 2019, Wang 2016). More broadly, the discussion still lacks an in-depth review of the processes specific to each model, which limits the ability to correctly attribute observed biases to their physical causes. In other words, this part remains superficial and would benefit from further development.

Thank you for specifically pointing to the 2 papers, which we revisited. The issue is now explained in more detail in the fifth paragraph of section 4.3.
We added further discussion to MIROC6 regarding their snow parameterizations. However, it is challenging to comprehensively relate all results to specific model features. It would need further sensitivity studies for each of the models. This statement was also added to the conclusions to highlight the need for further studies.

**For RC3:**

**Major revisions:**

- While the introduction has been significantly improved, it is still unclear how readers who are not familiar with the LS3MIP experiment will understand the objectives of the study.

   For example, the statement:

   _"the differences between the two can be used to attribute model biases to either the land surface model structure and parameterizations or coupled atmosphere-land interaction"_

   requires further clarification. How could a larger bias in the land-only model (compared to the coupled models) be attributed to parameterization issues or missing processes in the LSMs? If the same LSM is used, there is no obvious reason why the bias should be reduced in the CMIP6 experiment. This is explained in the LS3MIP experiment but not here.

   Similarly, the statement:

   _"Under identical, observation-based atmospheric conditions, the LS3MIP models are expected to simulate soil temperature more accurately than their CMIP6 counterparts."_

   requires additional justification. From a logical standpoint, this is not self-evident, even if modellers may agree that this is often the case in practice. A clearer explanation is needed to substantiate this claim.

We added information about LS3MIP to help readers understand what LS3MIP is, why and how we used it in this research. You can find the relevant changes in the last paragraph of the introduction.

- The discussion section in the revised manuscript remains difficult to follow and is not well structured.

   Several paragraphs contain sentences addressing multiple, unrelated aspects, which reduces readability. Furthermore, the section does not consistently compare the results with findings from previous studies, which would be essential for a proper scientific analysis.

   The section also suffers from inconsistencies in tense (switching between past and present), and it uses qualitative terms such as "better" or "good," which are not

scientifically precise.

While I have highlighted some of the most concerning sentences below, a thorough reorganization and rewriting of the discussion are necessary.

We went through and revised the discussion section and tried to make sure every statement is straightforward. We removed some repetitions and added more supporting points for our statements. We also addressed the tense and wording issues.

**Minor revisions:**

- Line 53 – Add a reference.

We added Koven et al. (2013) and Yokohata et al. (2020).

- Line 111 – The term _significant_ should not be used unless based on statistical analysis. Please be more precise.

Changed into „*Northern Eurasia contains more than two-thirds of the Earth's permafrost area (Groisman et al. 2007), with the majority located in Siberia.*"

- Lines 113–117 – These sentences could be shortened.

Revised as  „*Within this region, we used the soil temperature observational dataset at standardized depths, as provided by the All-Russian Scientific Research Institute of Hydrometeorological Information-World Data Center (RIHMI-WDC)* "

- Figure 5 – Align the orientation of the figure and its caption.

We simplified Figure 5 and put the previous to appendix (Figure A4), where we aligned the orientations accordingly.

- Figure 7 – In the authors' response, it is stated: _"The data have been grouped into intervals of 2°C, as illustrated in the histogram pairs. However, the x-axis ticks have been set at 5°C intervals in order to accommodate the wide range of temperatures and to avoid the cluttering."_ However, it remains unclear how a 4–6°C bin would be treated—would it belong to the 0–5°C or the 5–10°C interval? This should be clarified, as the current explanation is not consistent.

We thank the reviewer for pointing out this potential ambiguity. To clarify, the data is strictly grouped into 2°C intervals . The x-axis ticks are displayed every 5°C only for visual clarity and do not represent the binning. For example, a bin of 4–6°C is a distinct 2°C interval on its own, The center value of each bin can be seen by the center

x value of E5LC subplot and center x value of group pairs (between purple and green bars) in other subplots. We revised the figure caption to explicitly state this, so that the distinction between bin width (2°C) and axis tick marks (5°C) is unambiguous.

- Table 3 – Most readers may not be familiar with the meaning of _Kurtosis_. It would be useful to provide a short explanation in the discussion, including what it represents and why it is relevant in this context.
We added introduction of Kurtosis in Method section 2.5, and in discussion we further discussed about it.

- Section 4.1 – This section is not necessary or should not be part of the discussion.

Moved into method section 2.6.

- Line 531 – Rewrite this sentence for clarity.

Revised in the first of Section 4.1 and added more information.

- Line 545 – Avoid the use of _better_, as it is not scientifically precise.

 We revised the betters with clearer wording where necessary.

- Line 566 – The sentence _"Moreover, better snow simulation ability improved soil temperature simulation performance"_ needs clarification. Does this refer to results observed in this study? If so, where in the results is this demonstrated? If it is instead linked to the previous sentence regarding summer data, the connection is not evident and should not be made without further justification.

This general statement is actually not true for all models in our results, so we removed it.

- Line 568 – Provide a stronger motivation for this statement. How exactly does the data indicate deficiencies in representing surface energy exchange?

Rewrote as in the sixth paragraph of Section 4.2.

- Line 576 – The reference to Dutch et al. is problematic, since that study does not address soil insulation but only snow, and it focuses on a single site in Alaska outside the study domain.

After revising we no longer use this reference at this part of the article, but later and related only to snow.
Although its outside our domain, we still can use it in the context of improved parametrization for snow thermal conductivity, as the characteristic of frozen soil

snow should be similar.

- Lines 577–582 – This section begins with observational results (which should not appear here), then transitions into a speculative explanation involving thermal resistance or low soil moisture, which is insufficiently explained. This part requires substantial revision for clarity and coherence.

We have rewrote the discussion part. The relevant content could be found on the last paragraphs of Section4.2 and Section 4.3. And we removed redundancy of results in discussion.

- Line 597 – Since Dutch et al. is already cited for snow insulation earlier, consider consolidating the discussion here to avoid redundancy.

 We rewrote the contents according to this reference and now it is not repeating.

- Line 602 – While snow does indeed have low thermal conductivity, the statement should be made more specific. The value varies considerably by region and conditions.

We now include a range of values, from fresh to dense snow.

- Line 609 – This sentence is unclear. If it is merely a description of the figure, it should not be part of the discussion.

We rewrote it, including more detailed discussion (Section 4.3, third and forth paragraph).

- Lines 615–619 – Consider integrating relevant discussions from Dutch et al. and Damseaux et al. here.

Thank you for highlighting the relevance of these references for this part of our discussion. We included them, noting their primary finding that using the snow thermal conductivity parameterization described by Sturm et al. (1997) enhanced Arctic soil temperatures in CLM5.0.

- Line 649 – Clarify what is meant by _confirmed by the results_.

We removed this wording.

- Line 691 – This sentence requires clarification.

Rewrote as in Section 4.3, paragraph 5.

- Line 717 – This finding is also demonstrated in Damseaux et al. and could potentially be added here.

Yes, this supports our conclusion, we referred accordingly.

**Comment on the authors' response:**

- The classification of "Permafrost stations" remains problematic. The revised definition using the two-year criterion is an improvement, but the map still shows inconsistencies—for example, some northern locations are excluded while some more in the south are included. This likely results from the criterion requiring "at least 300 days of valid data every year." To avoid confusion, it would be clearer to rename these sites as "valid stations", or something else, rather than "permafrost stations," since the current label may lead readers to believe that excluded stations are not underlain by permafrost.

We now use the name ‚valid permafrost stations'.

- In the response, the authors stated: _"In the revised manuscript, we will provide a link to Zenodo where we will upload the relevant scripts."_ However, no such link is currently included. Please add this reference.

We have sorted the scripts and uploaded them. The Zenodo link is now provided in the manuscript.